# The SMN complex drives structural changes in human snRNAs to enable snRNP assembly

Josef Pánek [1,9] ✉, Adriana Roithová[2,8,9], Nenad Radivojević[2], Michal Sýkora [2], Archana Bairavasundaram Prusty [3], Nicholas Huston[4], Han Wan[5], Anna Marie Pyle [5,6,7], Utz Fischer [3] & David Staněk [2] ✉

Spliceosomal snRNPs are multicomponent particles that undergo a complex maturation pathway. Human Sm-class snRNAs are generated as 3′-end extended precursors, which are exported to the cytoplasm and assembled together with Sm proteins into core RNPs by the SMN complex. Here, we provide evidence that these pre-snRNA substrates contain compact, evolutionarily conserved secondary structures that overlap with the Sm binding site. These structural motifs in pre-snRNAs are predicted to interfere with Sm core assembly. We model structural rearrangements that lead to an open pre-snRNA conformation compatible with Sm protein interaction. The predicted rearrangement pathway is conserved in Metazoa and requires an external factor that initiates snRNA remodeling. We show that the essential helicase Gemin3, which is a component of the SMN complex, is crucial for snRNA structural rearrangements during snRNP maturation. The SMN complex thus facilitates ATP-driven structural changes in snRNAs that expose the Sm site and enable Sm protein binding.

Spliceosomal small nuclear RNAs (snRNAs) are key components of the spliceosome. They were discovered more than 40 years ago by Lerner and Steitz as RNA components of small nuclear ribonucleoprotein particles (snRNPs) co-precipitating with Sm proteins[1]. Intensive research during the following years revealed a complex biogenesis pathway that leads to formation of snRNPs and their essential role in RNA splicing (reviewed in refs. 2–6). All spliceosomal snRNAs (except U6 and U6atac) are synthesized by RNA polymerase II. The nascent snRNA is cleaved by the Integrator complex downstream of the mature 3′ end and released as a 3′ end extended precursor (pre-snRNA)[7]. Pre-snRNAs are exported to the cytoplasm bound to a protein complex containing XPO1 (CRM1), NCBP2 (CBP20)/NCBP1(CBP80), PHAX, and SRRT (ARS2), which interacts with 7-methyl-guanosine cap found at the 5′ end of pre-snRNAs[8–10]. In the cytoplasm, a ring from seven Sm

proteins (SNRPB/SmB/B′, SNRPD1/SmD1, SNRPD2/SmD2, SNRPD3/ SmD3, SNRPE/SmE, SNRPF/SmF and SNRPG/SmG) forms around a conserved single-stranded U-rich sequence in snRNAs, termed the Sm site. Formation of this so-called Sm core is facilitated by the multisubunit SMN complex acting in concert with the PRMT5 complex[11–14].

The cytoplasmic SMN complex consists of nine proteins named SMN, Gemin2-8, and Unrip[4]. A study analyzing the importance of individual SMN complex components revealed that SMN and Gemin2-4 are essential for Sm core assembly[15]. SMN and Gemin2 directly associate with Sm proteins bound to the assembly chaperone pICln and induce its release to enable Sm protein loading onto the Sm site[16,17]. Gemin5 directly binds to key features of snRNAs, the 5′ cap and the Sm site and hence may act as the "identifier" of the RNA substrate[18–21]. Gemin3 was shown to associate with factors important

[1]Laboratory of Bioinformatics, Institute of Microbiology, Czech Academy of Sciences, Prague, Czech Republic. [2]Laboratory of RNA Biology, Institute of Molecular Genetics, Czech Academy of Sciences, Prague, Czech Republic. [3]Department of Biochemistry, Theodor Boveri Institute, University of Würzburg, Würzburg, Germany. [4]Department of Molecular Biophysics & Biochemistry, Yale University, New Haven, USA. [5]Department of Molecular, Cellular, and Developmental Biology, Yale University, New Haven, USA. [6]Department of Chemistry, Yale University, New Haven, USA. [7]Howard Hughes Medical Institute, Chevy Chase, USA. [8]Present address: Laboratory of Regulation of Gene Expression, Institute of Microbiology, Czech Academy of Sciences, Prague, Czech Republic. [9]These authors contributed equally: Josef Pánek, Adriana Roithová. ✉e-mail: panek@biomed.cas.cz; stanek@img.cas.cz

for snRNP biogenesis but its role in Sm core assembly remains elusive[22]. Despite numerous studies describing molecular details of Sm ring formation there is an unresolved question regarding the ATP requirement. While formation of the Sm core on snRNA from purified Sm proteins does not require ATP[23–25], the assembly reaction performed in various cellular extracts is strictly ATP-dependent[12,14,26]. The ATP-dependent step in Sm core formation in cellular extracts has not yet been identified but ATP might be important for Gemin3, a putative ATP-dependent RNA helicase[27–29].

Formation of the Sm ring on snRNA is followed by methylation of the 5′ 7-methyl-guanosine cap to 2,2,7-trimethyl-guanosine and 3′ end trimming to produce the mature form of snRNA[4]. The Sm ring stabilizes snRNAs and is essential for transport of newly formed snRNPs back to the nucleus and to Cajal bodies[30–36]. In the Cajal body, several snRNA nucleotides are modified, snRNP-specific proteins are added and U4, U5, and U6 snRNPs are combined into the tri-snRNP[2,37–39].

SnRNAs have elaborate secondary and tertiary structures. The spatial organization of mature snRNAs in snRNPs and the spliceosome has been analyzed by chemical and enzymatic probing, and in recent years by X-ray crystallography and cryo-electron microscopy[3,40–55]. In all reported snRNA structures, the Sm binding site is always single-stranded and occupied by Sm proteins. However, the pre-snRNA export complex interacts with pre-snRNAs via the 5′ cap leaving the Sm site available for intramolecular base-pairing[8]. The structures of natural pre-snRNA substrates for the SMN complex can thus differ from the structures that snRNAs adopt in mature snRNPs.

To get insight into the structural landscape of pre-snRNAs we applied a combination of experimental and computational approaches. First, we predicted secondary structures of pre-snRNAs in several eukaryotic species and identified conserved compact secondary structures that would be expected to interfere with Sm protein binding. The accuracy of the predicted compact structures was evaluated with selective 2′ hydroxyl acylation analyzed by primer extension and mutational profiling (SHAPE-MaP) for human U2 and U4 snRNAs. We then computationally analyzed the structure rearrangement pathway that opens up the structured Sm site and makes it available for Sm proteins. The modeling suggested a two-step pathway that requires an extrinsic factor(s) for initiating the structural changes. We therefore applied selective knockdown of Gemin3, 4, and 5 and analyzed the effect on biogenesis of structured and unstructured snRNAs. Finally, in vitro studies provided evidence that Gemin3 is important for the ATP-driven structural rearrangement of U2 snRNA and U2 snRNP maturation.

## Results

### Structural prediction of pre-snRNAs reveals compact conserved structure involving the Sm motif

We decided to map the landscape of potential structures that human pre-snRNAs can adopt using comparative suboptimal secondary structure prediction. Unlike single minimum free energy (MFE) structures, we used multiple suboptimal secondary structures that can be predicted for a single RNA sequence with increasing free energy. Among these structures, a structure better corresponding to the native structure than the MFE structure is likely to exist, as snRNA structures with free energies higher than MFE are likely to correspond to the structures formed under native cellular conditions. To identify the best representative structure of human pre-snRNAs from a pool of multiple predicted suboptimal secondary structures we employed an evolutionary conservation criterion that we computed using pre-snRNAs of human and other metazoan species for which they were available (see Fig. 1a for a workflow and "Methods" for details).

Using this approach, we analyzed major snRNAs U1, U2, U4, U5 from 11 species representing various animals (for details see Supplementary information, Table S1). To comprehensively cover the structural space of individual pre-snRNAs, we chose evolutionarily distant

species from different branches of the phylogenetic classes. The available snRNA sequences were extracted from NCBI, Rfam and the archive of uRNADB public databases[56,57]. Because a single genome contains multiple gene copies of a given snRNA and some of them are incomplete or mutated, we removed all gene fragments shorter than 75% of the average length of the particular snRNA, filtered out snRNA sequences lacking the complete Sm site and, for U1 pre-snRNA, also lacking the U1-70K binding motif (for numbers of input sequences, see Supplementary information, Table S1). To model 3′ extension sequences, we mapped snRNA sequences from individual organisms to the corresponding genomes, extracted 3′ sequences downstream of the mature transcript and used the length of the 3′ extra sequence found in human pre-snRNAs[21] to model pre-snRNA sequences in selected organisms.

Then, we predicted suboptimal secondary structures for each pre-snRNA sequence using unconstrained prediction by UNAfold[58] with the exception of U1, to which we applied constrained prediction using RNAsubopt[59]. The unconstrained prediction by UNAfold did not provide consistent structures for U1 and we had to apply an additional constraint and blocked nucleotides (depicted by crosses in Fig. 1b) involved in the interaction with the SNRNP70 (U1-70K) protein from intramolecular base-pairing[52]. SNRNP70 was shown to interact with U1 pre-snRNA in the cytoplasm before or simultaneously with the SMN complex[60]. Therefore, it is rational to anticipate that SNRNP70 binding can affect the U1 pre-snRNA structure before snRNA is recognized by the SMN complex. We limited the number of suboptimal structures for a single sequence to 20, thus obtained 20 × number of sequences suboptimal structures for each pre-snRNA in each organism, among which we identified the most frequently occurring secondary structure and used this structure as the best representative of the pre-snRNA structure of the particular organism. Finally, we computed the structural conservation of the best representative structures based on their occurrence across all the selected animals. For each pre-snRNA, the structure with the highest conservation was identified as the most representative one. Examples of best representative structures of human pre-snRNAs are shown in Fig. 1b and for all analyzed species in Figs. S1–S4.

Various 3′ end extensions have been reported for various snRNA genes ranging from 5 to 49 nucleotides[21]. We therefore tested whether altering of the 3′ end extension affects folding of other pre-snRNAs, but we did not find any significant difference among structures with different lengths of 3′ end extra sequence for U2, U4, and U5 pre-snRNAs. In contrast, shortening of the 3′ extra sequence in U1 pre-snRNA sequence to only six extra nucleotides eliminated the compacted structure at the 3′ end and U1-pre-snRNA adopted a fold highly similar to the mature U1 snRNA structure, which indicates that U1 snRNA folding is specifically sensitive to 3′ extra sequence (Fig. 1c and Fig. S5)[52,61]. We were unable to predict consistent common best representative structures for minor U11, U12, and U4atac pre-snRNAs. Minor pre-snRNA structures showed significantly lower similarity of best representatives when compared to major pre-snRNAs (see Tables 1 and 2). However, significantly fewer sequences were available for calculation of the best representatives of minor snRNAs, which reduced the predictive power of our approach.

To test whether structural folding is conserved across eukaryotes, we applied the same approach to selected representatives of fungi (15 species) and protists (14 species) (Supplementary Information, Table S1). However, in contrast to Metazoa, we were unable to identify common best representative structures for any of the major pre-snRNAs indicating that suboptimal structures of fungi and protist pre-snRNAs lacked mutual structural similarity. The similarity of individual pre-snRNAs in each kingdom best representatives was significantly lower than the similarity of metazoan best representatives. To quantitatively determine structural differences, we compared pairwise

**a** Workflow to identify the most representative snRNA structure

**b** U1, U2, U4

**c** U1 alternative fold

**d** normalized tree edit distance

**U5**

- ● Sm site
- ● 3' pre-snRNA sequence
- ▬ Near Sm-site Structure (NSS)

structural distances of best representative suboptimal structures of metazoan with fungi and protist pre-snRNA homologs normalized to their sequence length (Fig. 1d). We also compared pairwise structure distances averaged for individual metazoan and protist/fungi major pre-snRNAs (Table 1). In both cases, two-sample t-test showed that the difference between metazoan and protist/fungi structural distances was statistically significant,

which indicated lower evolutionary conservation of protist and fungi pre-snRNA secondary structures. We also cannot fully exclude the possibility that the length of 3' end extra sequences in non-metazoan species are significantly different than in humans or that they are completely missing. We therefore decided to fully focus on human pre-snRNAs in the following experiments.

**Fig. 1 | In silico modeling of pre-snRNA secondary structures. a** Workflow of the computational procedure to identify best representative pre-snRNA secondary structures. For detailed description, see the Methods section. **b** In silico predicted structures of human pre-snRNAs. Best representative secondary structures for human pre-snRNAs containing 3′ extra sequences 49, 21, 7, and 48 nucleotides for U1, U2, U4, and U5 pre-snRNAs, respectively (based on ref. 21). **c** Alternative fold of human U1 pre-snRNA with a shorter 3′ end extension. Blue circles indicate extra 3′ end extension, red circles the Sm binding site and the beige lines the Near Sm-site Structure (NSS). **d** Box plot of the pairwise structural distances of the predicted secondary structures for Metazoa (M), fungi (F) and protists (P). Best representative structures of metazoan pre-snRNAs are significantly closer to each other (indicated by smaller normalized tree edit distances on y-axis) than those of fungi and protist species. On each box, the central mark indicates the median, and the bottom and top edges of the box indicate the 25th and 75th percentiles, respectively. The whiskers extend to the most extreme data points not considered outliers, and the outliers are plotted individually using the '+' symbol. *P*-values of the two-way Student *t*-test are indicated in Table 1. *n*−number of species assayed.

## SHAPE-MaP mapping of U2 and U4 snRNA secondary structure is consistent with compact structure around the Sm site

The predicted human pre-snRNA structures (Fig. 1b) differed from generally accepted structures of mature snRNAs derived from chemical and enzymatic probing[62] and lately from cryo-EM spliceosome structures[49,51,61,63,64]. The most striking difference were structured regions formed around the Sm site, which we termed the Near Sm-site Structure (NSS). In some cases, a few nucleotides of the Sm site were also involved in NSS formation. These compact structures might interfere with snRNP biogenesis and Sm ring formation because the Sm site is always single-stranded in snRNPs, with Sm proteins wrapped around it[52,62,65,66].

To test this model experimentally, we first employed SHAPE-MaP to compare the natural U2 snRNA structure in U2 snRNP (in vivo) with deproteinized cellular U2 snRNA (ex vivo) and U2 snRNA synthesized in vitro. We took advantage that 3′ end extension of U2 pre-

### Table 1 | Structural similarities among metazoan, fungi and protist pre-snRNA

| pre-snRNA | Metazoa | Protists* | Fungi** |
|---|---|---|---|
| pre2-u1 | 0.38 | 0.58/0.51 | 0.51/0.52 |
| pre2-u2 | 0.29 | 0.44/0.42 | 0.49/0.43 |
| pre2-u4 | 0.22 | 0.48/0.47 | 0.57/0.57 |
| pre5-u5 | 0.37 | 0.46/0.42 | 0.43/0.45 |

*Species best distributed across protist taxonomy space/species best distributed across taxonomy space with model organisms.

**Species best distributed across fungi phyla/species best distributed across fungi phyla with model organisms.

Pairwise similarities of best representative suboptimal structures averaged for individual metazoan, protist, and fungi major pre-snRNAs. Note that the less is the number, the higher is the structure similarity, as the similarity was computed using tree edit distances. Longest extra sequences were used to obtain pre-snRNA sequences. Two-sample two-tail t-test compared metazoan and protist similarities for alternative species distribution (see the table comments) with a *p*-value = 0.0115/0.0180, and metazoan and protist similarities with a *p*-value = 0.0079/0.0115. These *p*-values show that metazoan and protist, and metazoan and fungi populations have unequal means, which indicated that if metazoan best representative suboptimal structures showed structure similarity, protist/fungi best representative suboptimal structures should have structure dissimilarity, suggesting the NSS is not evolutionarily conserved in protist and fungi structures.

### Table 2 | Structural similarities among metazoan minor pre-snRNAs

| | |
|---|---|
| pre-u11 | 0.63 |
| pre-u12 | 0.58 |
| pre-u4atac | 0.48 |

Pairwise similarities of best representative suboptimal structures averaged for individual metazoan minor pre-snRNAs. Note that the less is the number, the higher is the structure similarity, as the similarity was computed using tree edit distances. Two-sample two-tail t-test compared metazoan major (see Table 1) and minor similarities with a *p*-value = 0.0077, showing that their dissimilarity was significant which indicated that if metazoan major had structure similarity, metazoan minor should have structure dissimilarity.

Statistically comparison of protists and fungi major (see Table 1) and metazoan minor similarities yielded a *p*-value = 0.0621, showing that their dissimilarity was insignificant which indicated that if fungi an protist major had structure dissimilarity, metazoan minor should have also structure dissimilarity.

snRNA does not affect formation of NSS to overcome the fact that cells contain very little pre-snRNA relative to the mature form. We applied the SHAPE reagent 2-methylnicotinic acid imidazolide (NAI), which preferentially modifies the 2′ OH groups of flexible nucleotides[67]. These bulky adducts are encoded as cDNA mutations when reverse transcription is performed in the presence of a manganese, instead of magnesium, cofactor. After reverse transcription, next-generation sequencing libraries are generated and sequenced to calculate mutation rates, which are subsequently converted into chemical reactivities using the ShapeMapper analysis pipeline and mapped on the published structure of U2 snRNA in snRNP (in vivo)[44] where sequences around the Sm site are single-stranded. It should be noted that the nucleotides of the Sm motif are not reactive under in vivo conditions due to protection by Sm proteins, as protein occupancy is known to inhibit reactivity with NAI[67]. The 2′ OH reactivities of in vitro and ex vivo snRNAs were used as constraints for the prediction of a secondary structure using RNAFold (Fig. 2a). These structures matched almost perfectly to in silico predicted U2 snRNA structures and are consistent with the presence of NSS (compare Figs. 1b and 2a). A comparison of ex vivo and in vivo SHAPE reactivities, as computed with the ΔSHAPE approach, revealed that SHAPE reactivities of the Sm site in the U2 snRNA are significantly higher ex vivo when compared to reactivities collected in vivo, which is consistent with Sm proteins protecting the Sm site in snRNP (Fig. 2b). In addition, in vivo reactivities for NSS are significantly elevated relative to ex vivo, suggesting that Sm protein binding is incompatible with NSS duplex formation (Fig. 2b). Although we experimentally analyzed the secondary structure of mature U2 snRNA, SHAPE-MaP results are in good agreement with in silico predictions and show that U2 snRNA stripped of proteins can adopt a different fold from that in U2 snRNP. The major difference lies in the central part of the molecule where nucleotides 40–111 form a long stem II (as denoted in Fig. 2a, ex vivo) in naked U2 snRNAs while in the mature fold, these nucleotides are mostly single-stranded with exception of two short helixes IIa and IIb between nucleotides 48–84 (Fig. 2a, in vivo).

Next, we probed a structure of U4 pre-snRNA synthesized in vitro. We did not analyze U4 snRNA isolated from cells because the majority of U4 snRNAs are base-paired with U6 snRNA, which would interfere with our measurements. We applied the SHAPE-MaP assay to identify single-stranded nucleotides chemically modified by NAI. The modified nucleotides were then used as constraints for the prediction of a secondary structure using SuperFold (Fig. 2c). This U4 pre-snRNA fold is almost identical to the in silico predicted structure (compare Figs. 1b and 2c). In general, SHAPE-MaP mapping results are consistent with the model that naked U2 and U4 snRNA folding deviated from published structures and sequences around the Sm site can adopt alternative more compact conformations.

## Structural context of the Sm site affects U2 snRNP biogenesis

Next, we tested whether structures around the Sm site can affect the biogenesis of U2 snRNP. We designed two mutants of human U2 snRNA that destabilize NSS (Fig. 3a, weakNSS and noNSS) and one mutant that strengthens the NSS helix (Fig. 3a, stNSS). The structures

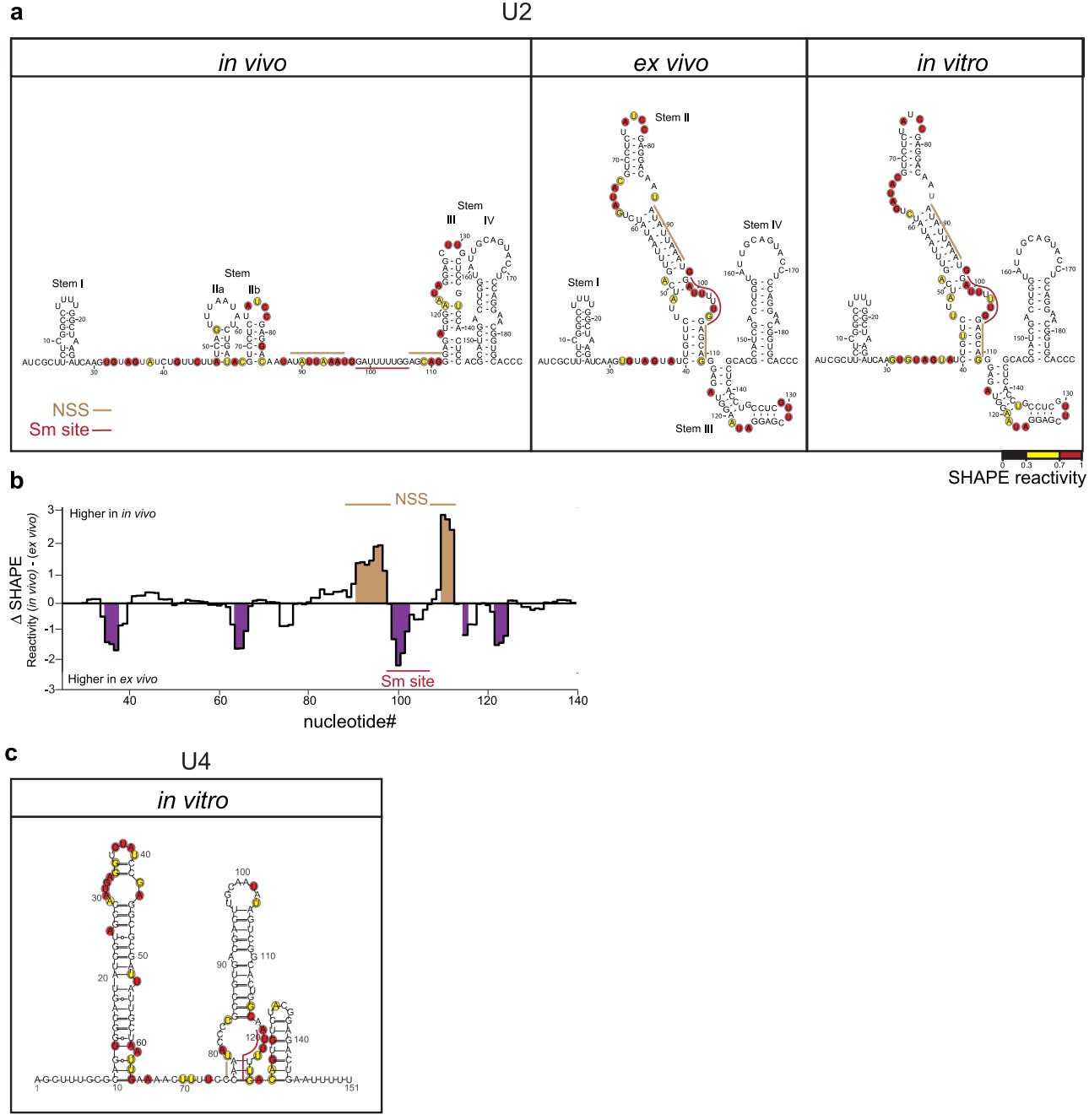

**Fig. 2 | Experimental determination of the U2 and U4 snRNA structures.**
**a** Secondary structure of U2 snRNA probed by SHAPE-MaP inside cells (in vivo), after isolation of cellular RNA (ex vivo) and transcribed in vitro (in vitro). Nucleotides reacting with the SHAPE reagent, indicated by red and yellow circles, were mapped onto the mature snRNA structure from U2 snRNP (based on ref. 44; in vivo) or used as constraints to model RNA structure (ex vivo and in vitro). **b** Comparison of the SHAPE reagent reactivity collected either in vivo or ex vivo for the U2 snRNA. Near Sm site Structure (NSS) predicted by in silico modeling (see Fig. 1) is marked by beige lines, the Sm site by a red line. Analysis of statistically significant reactivity differences between ex vivo and in vivo-determined SHAPE reactivities was performed using the DeltaSHAPE automated analysis tool and default settings. Shaded regions indicate regions in which SHAPE reactivities are significantly different between the two experiments (standard score ≥1). **c** Secondary structure of in vitro synthesized U4 pre-snRNA probed by SHAPE-MaP. Nucleotide reactivities with the SHAPE reagent, marked by red and yellow circles, were used to predict the presented structure.

of mutated U2 snRNAs were predicted as MFE structures (using RNA Vienna package[59]) with the structure of the best representative serving as a structural template. To test how the NSS structure affects the U2 snRNA biogenesis pathway, we microinjected mutated U2 variants into the cytoplasm of HeLa cells. We have shown previously that microinjected snRNAs enter the biogenesis pathway, acquire the Sm ring, and are transported to the nucleus and Cajal bodies[30,32,33,68]. U2 snRNAs were in vitro synthesized in the presence of Alexa488-UTP

and injected into the cytoplasm of HeLa cells. After 1 h incubation, cells were fixed and Cajal body localization of U2 variants was assayed (Fig. 3b). WT U2 snRNA properly localized to Cajal bodies, as shown before[30]. Similarly, relaxation of NSS did not affect the snRNA ability to enter the cell nucleus and Cajal bodies. However, compact NSS in U2stNSS, where all nucleotides of the Sm site are base-paired, reduced nuclear import and Cajal body accumulation. Because Cajal body targeting is mediated by Sm proteins[30], this finding indicates

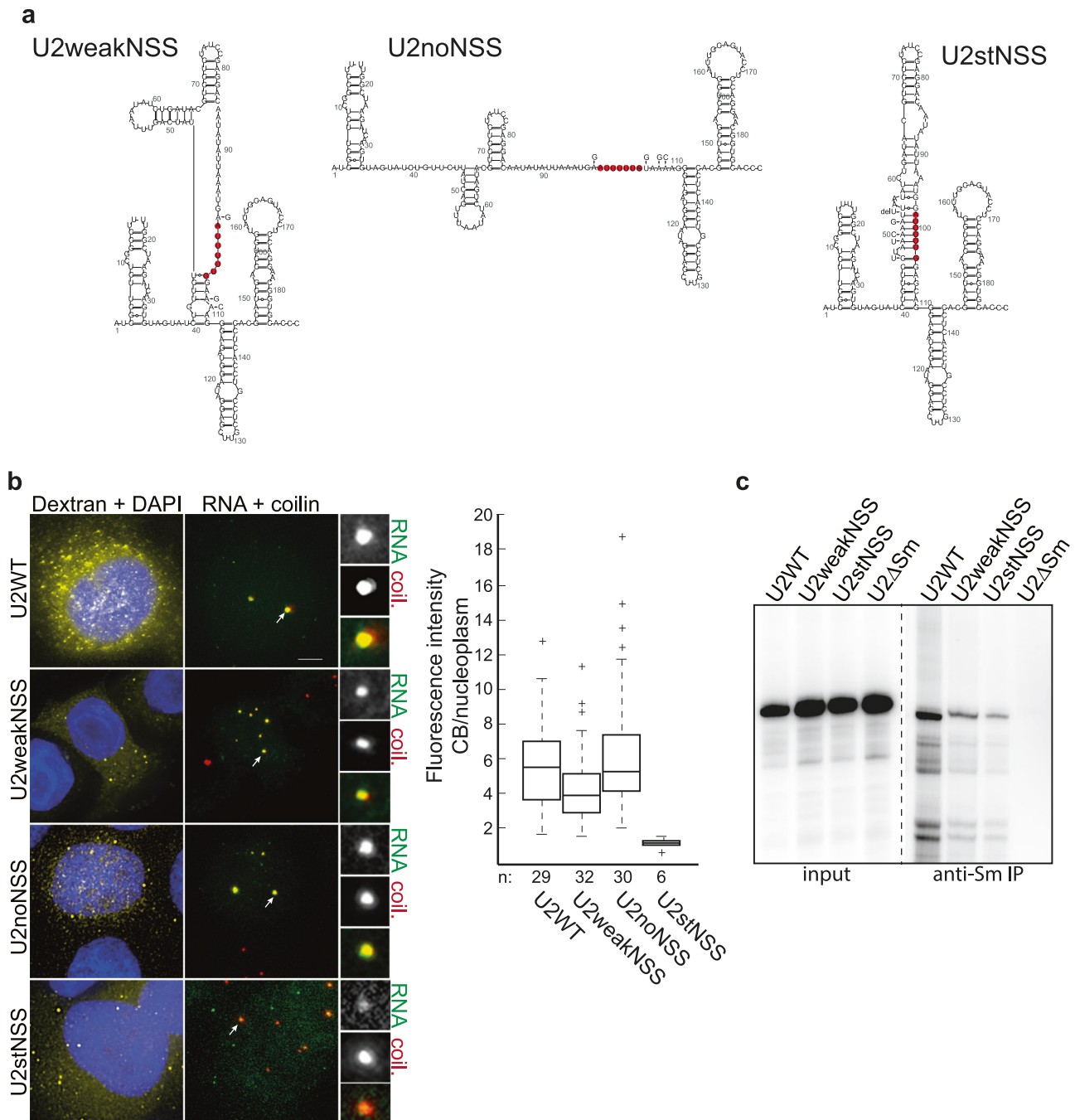

**Fig. 3 | The structure of NSS affects U2 snRNA biogenesis. a** Two mutants with weakened NSS and one mutant with enhanced base-pairing of the Sm site were designed, and the predicted impact of mutations on the structure of U2 snRNA is shown. **b** U2 snRNAs were transcribed in vitro and microinjected into the cytoplasm of HeLa cells. Cajal bodies were visualized by coilin immunostaining, and localization of snRNAs in Cajal bodies was monitored. Left panel: yellow−TRITC-dextran-70kD used as a marker of injection; blue−DAPI. Right panel: red−coilin, green−snRNA. Cajal bodies marked by arrows were enlarged 3 times and shown in insets. Scale bar represents 5 μm. Fluorescence intensities of injected snRNAs in Cajal bodies were normalized to the signal in the nucleoplasm and are shown in the box plot. *n*−number of cells assayed. On each box blot, the central mark indicates the median, and the bottom and top edges of the box indicate the 25th and 75th percentiles, respectively. The whiskers extend to the most extreme data points not considered outliers, and the outliers are plotted individually using the '+' symbol. **c** SMN-assisted assembly of Sm proteins on individual U2 snRNA mutants. Radioactively labeled in vitro transcribed snRNAs were incubated with purified SMN complexes. Interaction of snRNAs with Sm proteins was monitored by anti-Sm immunoprecipitation followed by autoradiography of co-precipitated snRNAs.

that base-pairing of the Sm site inhibits formation of the Sm ring. To explore this possibility, we incubated in vitro transcribed U2 snRNAs with a purified SMN complex and analyzed Sm ring formation by snRNA co-immunoprecipitation with Sm proteins (Fig. 3c). Consistent with the microinjection experiment, formation of the Sm ring on the U2stNSS substrate was reduced in comparison to WT and weakNSS U2 constructs. In vitro data further suggest that NSS presence might enhance SMN-dependent Sm ring assembly because we observed lower Sm protein association with weakNSS construct than WT construct. These results show that the structural context of the Sm site is important for U2 snRNA biogenesis and Sm ring assembly.

## Opening up the compact secondary structure requires an extrinsic factor

Mathematical modeling and SHAPE-MaP analysis were both consistent with the model that the U2 pre-snRNA containing the NSS, which we termed the "primary folded transcript" in the following text, is a natural substrate for the SMN complex. We also provided evidence that structural context of Sm site influences snRNP biogenesis. We therefore decided to apply in silico modeling to get insight into structural rearrangements that would allow NSS opening and adoption of the mature snRNA with a single-stranded Sm site accessible for interaction with Sm proteins. We did not identify a direct single-step pathway that would lead to the desired rearrangements of primary folded transcripts, and a two-step restructuring process had to be applied to achieve that (see Fig. 4a for a computing workflow).

We started our analysis with the predicted best representative secondary structures (shown in Fig. 1b and Figs. S1–S4). In the first step, NSS opening was achieved using constrained refolding of the best representative structures, when the nucleotides forming NSS were blocked from base-pairing. To mimic the putative real restructuring process, the refolding constraint combined both the best representative structure and the blocked nucleotides forming NSS (see Supplementary Data 1 for a detailed description of refolding constraints). We identified the minimum number of blocked pairs in NSS in the direction from its root to its loop for each pre-snRNA (shown as dashed line in Figs. 4b and S6 for human pre-snRNAs, and for the remaining evaluated metazoan species in Supplementary Data 1 in dot-bracket form). Blocking more base pairs toward the NSS loop resulted in the same effect. The folding algorithm not only unpaired blocked nucleotides, but also unfolded other base-paired parts of NSS to keep the predicted structure thermodynamically valid. The constrained unfolding rendered the Sm site single-stranded, while the rest of the structure remained intact (Figs. 4b and S6 for human snRNAs, Figs. S6–S10 for all evaluated metazoan species). Interestingly, NSS, Sm site, and surrounding sequences occupy thermodynamically one of the least stable parts of the primary folded transcripts thanks to prevailing A-U and G-U base-pairing formed due to multiple uridines both within and around the Sm motif (Fig. S11). We termed the structures obtained by the rearrangement of primary folded transcripts "folding intermediates".

To reach the final structure, we forced nucleotides of the Sm motif and a few nucleotides downstream of the Sm motif to remain single-stranded to mimic bound Sm proteins (Figs. 4b and S6 for human pre-snRNAs, and in Supplementary Data 2 for the remaining evaluated metazoan species in dot-bracket form). This way, folding intermediates refolded into mature structures (Figs. 4b and S6 for human pre-snRNAs and Figs. S12–S15 for all evaluated metazoan species). The same two-step remodeling procedure allowed us to predict the folding pathway for all major pre-snRNAs.

## Gemin3 is important for ATP-driven U2 snRNA rearrangement during snRNP biogenesis

In a next step, we tested the model of NSS opening experimentally. The SMN complex contains two potential candidates that can induce NSS unwinding. Gemin5 has been shown to interact with the 7mG cap, the Sm motif, and a stem at the 3′ end[19–21]. A few uridines from the Sm site always bulge out from NSS in primary folding transcripts (Fig. 1), which makes them accessible for Gemin5 binding, which can induce NSS melting. Alternatively, the essential DEAD-box helicase Gemin3 (DDX20) with unknown function may partially open up NSS. Gemin3 firmly interacts with another essential SMN component Gemin4, which likely regulates its activity[69,70]. To test the effect of these proteins on snRNP biogenesis, we knocked down Gemin3, 4, and 5 by RNA interference, microinjected fluorescently labeled WT U2 snRNA into the cytoplasm and monitored its accumulation in the Cajal body. While Gemin5 downregulation did not have any effect on Cajal body

localization, depletion of Gemin3 and to a lesser extend also depletion of Gemin4 reduced nuclear import and Cajal body accumulation of WT U2 snRNA (Fig. 5a).

If Gemin3 is essential for opening up the NSS structure, then artificial NSS relaxation should remove the Gemin3 requirements. To test this prediction, we either heat-denatured WT U2 snRNA before injection or injected the U2noNSS construct (Fig. 5b). In both cases, snRNAs localized to the Cajal body independently of the Gemin3 presence. To further test the role of Gemin3, we inserted the MS2 binding loop into WT and weakNSS U2 snRNAs and co-expressed these constructs with MS2-YFP. Under physiological conditions, both constructs localized to Cajal bodies, as described previously[30] (Fig. 5c). However, Gemin3 downregulation reduced Cajal body localization of the WT U2 construct but not U2weakNSS, which is consistent with a function of Gemin3 in NSS relaxation.

Next, we monitored whether the nuclear import and Cajal body localization of U1, U4, and U5 pre-snRNAs also depends on Gemin3. We did not observe any nuclear import of U5 pre-snRNA injected into the cytoplasm of control cells and did not analyze U5 snRNA any further (Fig. S16a). Similarly, the U1 pre-snRNA with the 3′ extra sequence found downstream of the main *RNU1-1* gene did not localize to the nucleus and Cajal bodies (Fig. S16b). We therefore tested an alternative 3′ extra sequence from the *RNU1-26P* gene. The alternative extra 3′ sequence supported a different structural fold that was however also commonly found among the best representatives of human U1 pre-snRNA (Fig. S16c). Similarly to U2 snRNA, U1 and U4 pre-snRNAs localization to the Cajal body was dependent on Gemin3 (Fig. 6a, b). Next, we introduced point mutations that either tightened (U1/U4stNSS) or relaxed (U1/U4noNSS) structured regions around the U1 and U4 Sm sites (Fig. S16c, d). Consistently with U2 microinjection data, compacted structure in stNSS constructs prevented Cajal body accumulation while relaxation of NSS removed the requirement for Gemin3 (Fig. 6a, b).

Next, we analyzed the effect of Gemin3 on in vitro Sm ring formation using U2 snRNA as a template. We incubated radioactively labeled U2 snRNAs in cytoplasmic extracts prepared from cells treated with anti-Gemin3 siRNA and monitored Sm ring assembly by anti-Sm immunoprecipitation (Fig. 7a). The Sm ring was formed on WT snRNA and its assembly was inhibited by Gemin3 knockdown. In contrast, binding of Sm proteins to U2noNSS was only slightly reduced upon Gemin3 knockdown. Stabilization of NSS (U2stNSS) completely inhibited Sm protein association, which indicates that conditions in the cell extract are more stringent than conditions when the isolated SMN complex is used for Sm ring assembly (compare results for U2stNSS constructs in Figs. 3c and 7a).

These data strongly suggest that Gemin3 role in snRNP biogenesis involves relaxation of NSS. To test this prediction experimentally, we prepared a molecular beacon that mimics U2 NSS where the 3′ end was labeled with Texas red and the 5′ end with a fluorescence quencher. When the molecular beacon is in the folded NSS-like conformation, the quencher reduces Texas red fluorescence, while relaxation of NSS leads to increased Texas red fluorescence. Incubation of the molecular beacon with the SMN complex increased Texas red fluorescence, indicating relaxation of the NSS (Fig. 7b). Preincubation of the SMN complex with non-hydrolyzable ATP analog γ-S-ATP reduced fluorescence signal when compared to SMN complex incubated in a buffer alone. This suggests that binding of the SMN complex containing γ-S-ATP fixes NSS and keeps the 5′ and 3′ ends close to each other. We surprisingly observed NSS opening even without addition of external ATP. One explanation could be that the isolated SMN complex contains residual pre-bound ATP, which can be utilized during the reaction. These data together show that the SMN complex has ATP-dependent NSS relaxation activity.

Finally, to assay whether Gemin3 is involved in relaxation of U2 NSS, we prepared a HeLa cell line where the *DDX20* gene (Gemin3)

**a**

Workflow to model structural changes during snRNA biogenesis

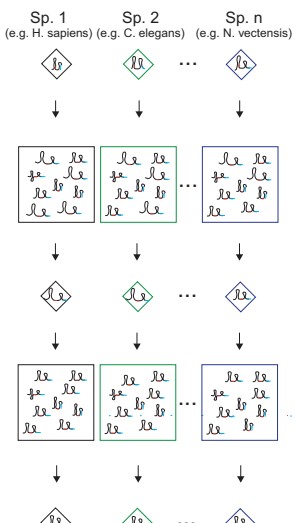

1. Primary folded transcripts (best representative suboptimal structures, one structure for each species)

*2. Predict suboptimal structures by RNAsubopt with manually set constraint: "A minimal number of nucleotides at NSS root blocked from base pairing." This constraint unfolds NSS and surrounding structures.*

3. Sets of subotimal structures for folding intermediates with relaxed NSS structure (one set of each species)

*4. Find suboptimal structures (one structure from each set, i.e. from each species) that are most mutually similar to each other among suboptimal structures across all sets, i.e. species*

5. Folding intermediates

*6. Predict suboptimal structures by RNAsubopt with manually set constraint: "Nucleotides blocked from base pairing due to interaction with Sm proteins."*

7. Sets of suboptimal structures for final structures, one set for each species

*8. Find suboptimal structures (one structure from each set, i.e. from each species) that are most mutually similar to each other among suboptimal structures across all sets, i.e. species*

9. Final structures

**b**

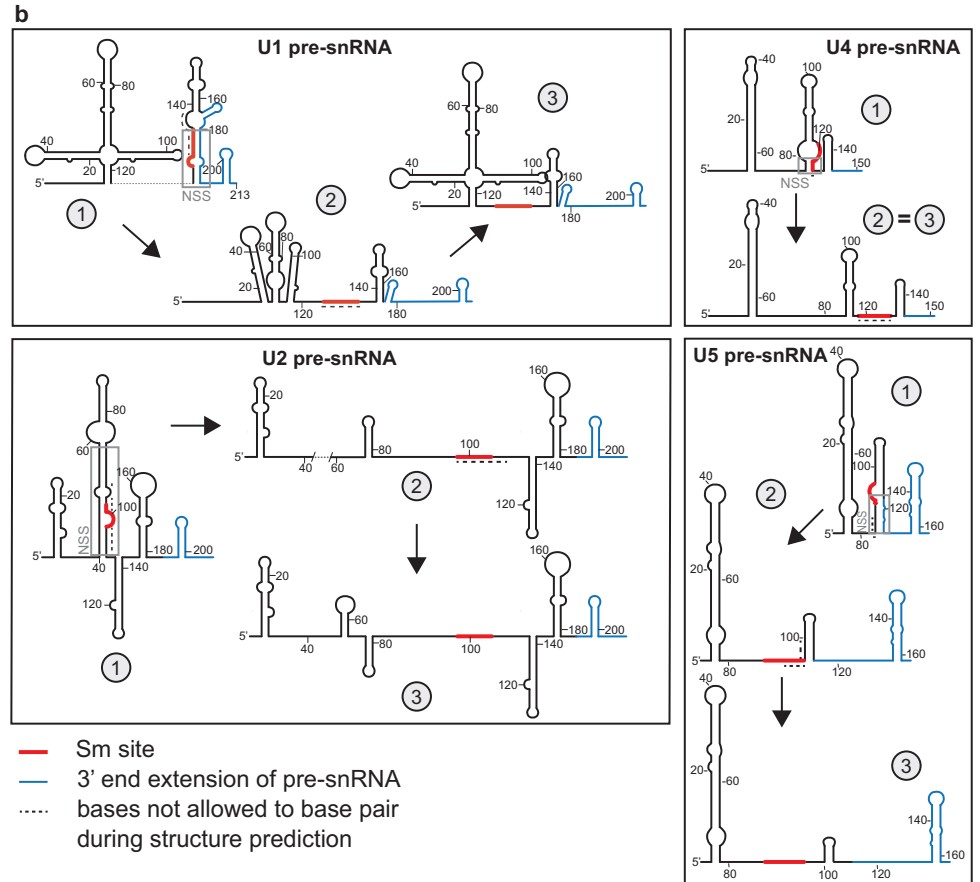

Sm site
3' end extension of pre-snRNA
bases not allowed to base pair during structure prediction

**Fig. 4 | In silico predicted structural changes during pre-snRNA biogenesis.**
**a** Workflow of the computational procedure to model changes of pre-snRNA secondary structures. For detailed description, see the "Methods" section. **b** Schematic representation of a two-step pathway that describes structural changes during biogenesis of human pre-snRNAs from the primary folded transcript (1), via folding intermediates (2) to the final structure (3). Nucleotides that were prevented from base-pairing are indicated by a dotted line. Note that in the case of U4 pre-snRNA, only one nucleotide at the root of NSS was blocked from base-pairing in the first step. U4 pre-snRNA adopted the final structure fold after first structural rearrangements and no structural changes were observed between the folding intermediate and the final structure. NSS is marked by a gray box, the Sm site and 3' extra sequence by red and blue line, respectively.

was conjugated with a tag containing EGFP and the FKBP12[F36V]-based degron (Fig. S17a)[71]. The 72 h treatment with dTAG13, which activated the degron effectively reduced the amount of Gemin3 (Fig. S17b, c). We then purified the SMN complexes from mock treated and dTAG13 treated cells. We observed that Gemin4 and Gemin5 proteins were depleted along with Gemin3 from the complex while the stochiometric ratio of other SMN complex components remained unchanged (Fig. S17d), which is consistent with previous findings[72]. Then we

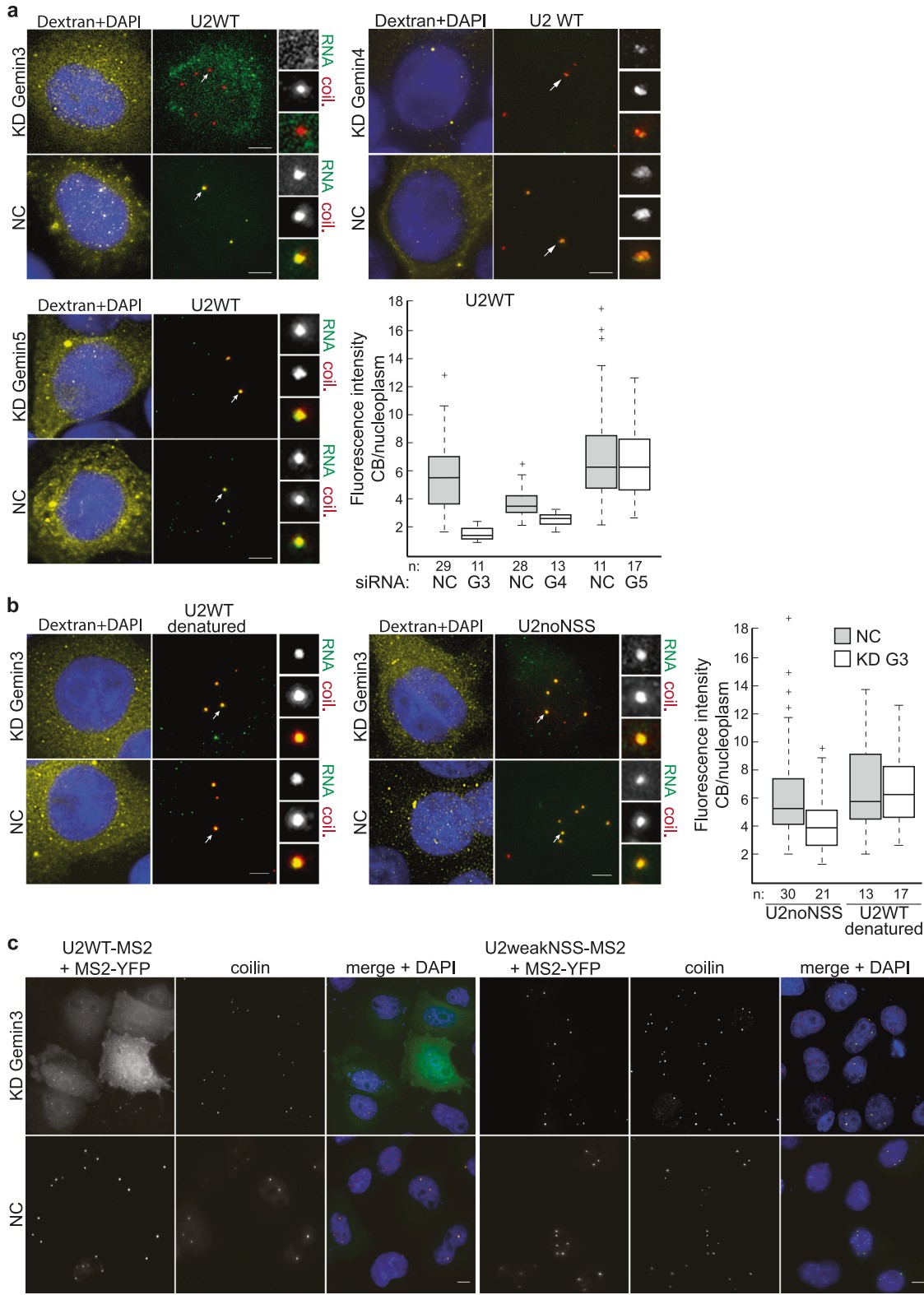

incubated the U2-mimicking molecular beacon with purified SMN complexes. In both cases, we observed partial increase of Texas red fluorescence indicating relaxation of NSS. The fluorescence induction was less pronounced when the molecular beacon was treated with the SMN complex depleted of Gemin3 (and Gemin4 and 5). These data suggest that Gemin3 together with Gemin4 and 5 are important factors that restructure the snRNA template to allow Sm ring formation.

## Discussion

In silico modeling of pre-snRNA secondary structures suggested a previously unidentified fold for all major pre-snRNAs transcribed by RNA polymerase II (U1, U2, U4, U5) (Fig. 1) that is characterized by evolutionarily conserved secondary structure involving sequences around and within the Sm motif (Figs. S1–S5). It should be mentioned that pre-snRNA sequences from non-human species were not experimentally validated. However, the fact that we were able to identify

**Fig. 5 | Gemin3 is important for U2 snRNA localization in Cajal bodies. a** Gemin3, Gemin4 and Gemin5 were downregulated by RNAi, fluorescently labeled WT U2 snRNA was microinjected into the cytoplasm. G3–Gemin3, G4–Gemin4, G5–Gemin5, NC–negative control siRNAs. The box plot shows minimum/maximal values (whiskers), first/third quartile values (rectangle), median (solid line), and outliers (+). **b** Relaxation of U2 snRNA structure by denaturation (U2WT denatured) or mutations (U2noNSS) removed requirements for Gemin3, and both U2 snRNAs reached Cajal bodies in the absence of Gemin3. Cajal bodies marked by arrows were enlarged 3 times and shown in insets. Scale bar represents 5 μm. Localization of injected snRNAs in Cajal bodies (marked by coilin immunostaining) was monitored, quantified and shown in box plots. *n*–number of cells assayed. On the box plots in (**a**) and (**b**), the central mark indicates the median, and the bottom and top edges of the box indicate the 25th and 75th percentiles, respectively. The whiskers extend to the most extreme data points not considered outliers, and the outliers are plotted individually using the '+' symbol. **c** WT U2 snRNA containing the MS2 loop (U2WT-MS2) or U2 snRNA containing mutations weakening the NSS structure and the MS2 loop (U2weakNSS-MS2) were co-expressed with MS2-binding protein tagged with YFP (MS2-YFP) in cells treated with negative control siRNA (NC) or cells treated with anti-Gemin3 siRNA (KD Gemin3). U2weakNSS-MS2 localized to Cajal bodies (marked by coilin immunostaining) in both cases, while U2WT-MS2 only in control siRNA-treated cells. MS2-YFP–green, coilin–red, and DAPI–blue. Scale bar represents 10 μm.

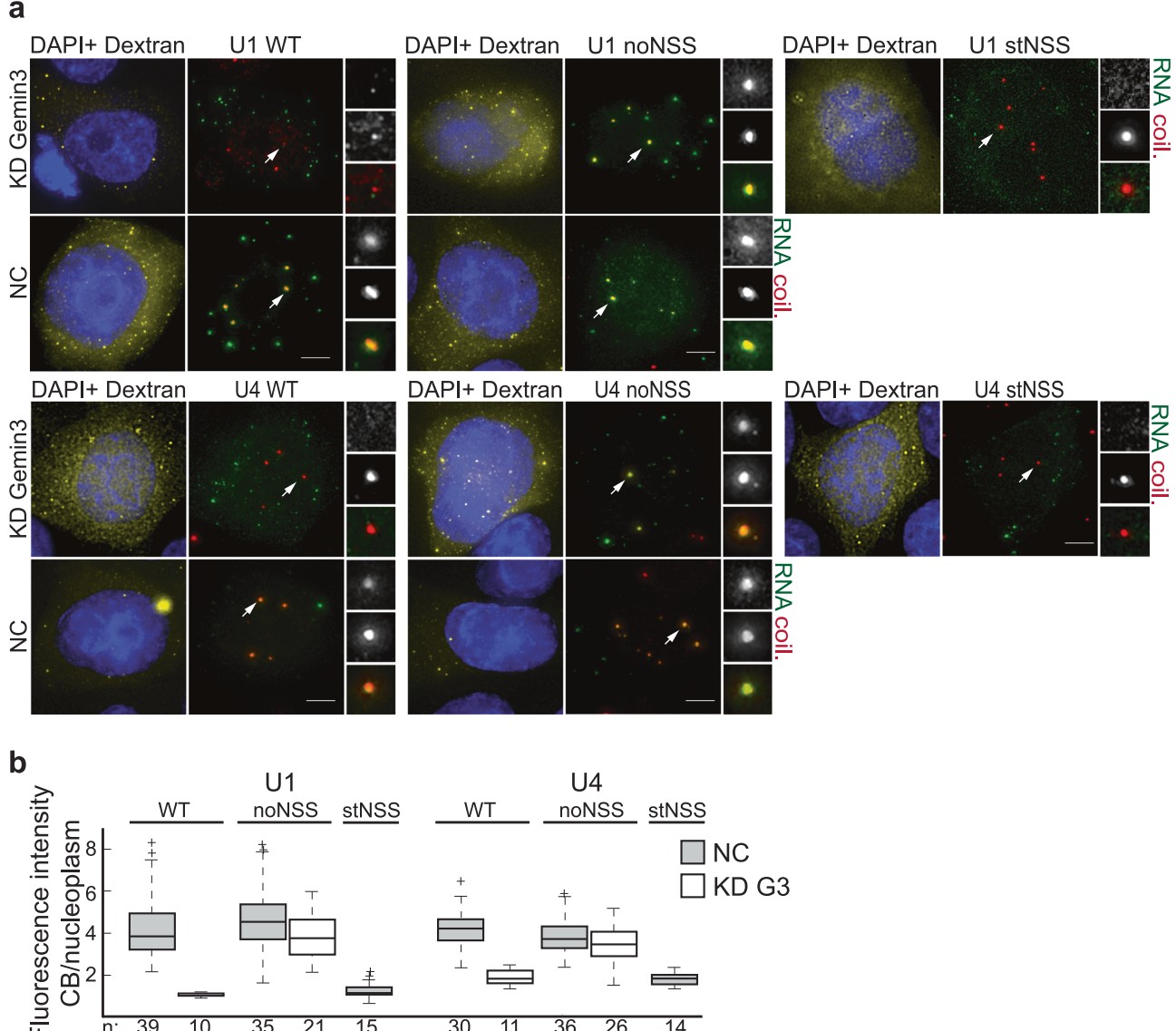

**Fig. 6 | Gemin3 is important for U1 and U4 localization in Cajal bodies. a, b** Gemin3 was downregulated by RNAi and fluorescently labeled U1 and U4 pre-snRNAs were microinjected into the cytoplasm. Pre-snRNAs with relaxed NSS (noNSS) did not require Gemin3 to reach Cajal bodies. Strengthening NSS prevented Cajal body localization even in cells treated with negative control siRNA. **a** Cajal bodies marked by arrows were enlarged 3 times and shown in insets. Scale bar represents 5 μm. **b** Localization of injected snRNAs in Cajal bodies (marked by coilin immunostaining) was monitored, quantified, and shown in box plots. *n*–number of assayed cells. On each box, the central mark indicates the median, and the bottom and top edges of the box indicate the 25th and 75th percentiles, respectively. The whiskers extend to the most extreme data points not considered outliers, and the outliers are plotted individually using the '+' symbol.

common structures for all animal species indicates that 3′ extended snRNA precursors likely exist also in other animals than humans. If it was not the case, the structural similarity could not be computationally detected, because the extra sequences often participate on forming secondary structures of pre-snRNAs. The existence of such secondary structural motifs was demonstrated experimentally for in vitro transcribed human U2 snRNA and U4 pre-snRNA with SHAPE-MaP analysis (Fig. 2). In the case of U2 snRNA, the most striking structural difference

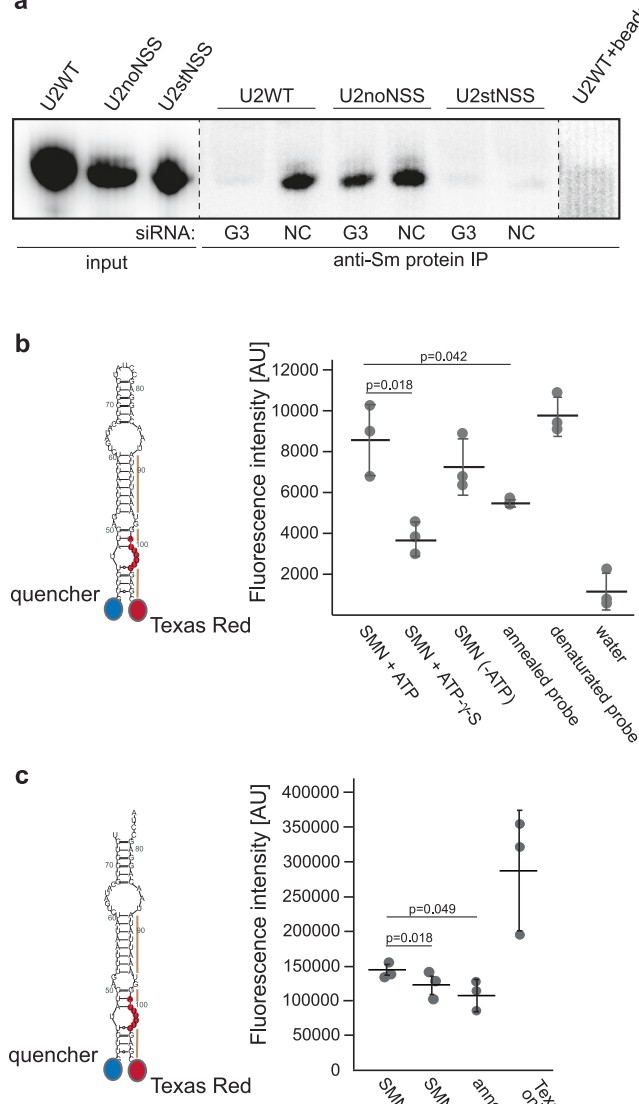

**Fig. 7 | Gemin3 is important for Sm protein loading on U2 snRNA. a** In vitro transcribed U2 snRNA (WT and mutants) was radioactively labeled and incubated in cytoplasmic extracts prepared from control cells or anti-Gemin3 siRNA-treated cells. Sm protein association with snRNA was assayed by immunoprecipitation of Sm proteins followed by autoradiography of co-precipitated snRNAs. **b** A molecular beacon mimicking the hairpin structure of U2 NSS was incubated with a purified SMN complex with or without external ATP or with the non-hydrolyzable ATP-γ-S analog. The denatured molecular beacon served as a positive control, incubation in PBS as a negative control. Three independent experiments were performed, mean and SEM are shown. **c** molecular beacon mimicking the hairpin structure of U2 NSS was incubated with a purified SMN complex treated with DMSO (negative control) or dTAG13 activating the degron in the Gemin3-GFP-FKBP12$^{F36V}$ protein. Annealed probes served as negative control and Texas red probe only as a positive control. Three independent experiments were performed, mean and SEM are shown. Statistical significance was tested by paired, two-tail t-test.

with respect to the mature structure folded in snRNP involves nucleotides between 40 and 111 that form a long stem in deproteinized U2 snRNA. In the mature U2 snRNA structure in the U2 snRNP, this sequence is mostly single-stranded except for two short stem loops IIa and IIb. Stem loops I, III, and IV are stable and their folding is similar in both structures regardless of rearrangements in the central part of the

molecule. An alternative structure for the central part of *Drosophila* U2 snRNA termed stem loop II was proposed by Keller and Noon[73], and this structure is similar to the top of stem loop II in our prediction (nucleotides 53–95). Later on, Ares and Igel tested the stem loop II structure in yeast by extensive mutagenesis and concluded that the stem loop II structure breaks in two shorter stem loops termed IIa and IIb[42]. With a few exceptions (e.g., ref. [74]), the structure involving stem loops IIa and IIb was since then accepted as the major structure of the mature U2 snRNA.

Later studies identified rearrangements of stem loops IIa and IIb during the splicing reaction[75,76]. Thus, the folding of the central sequence of U2 snRNA seems flexible and can adopt alternative structures based on the environment and bound proteins. Consistent with this structural flexibility we detected substantial rearrangement of stem loops IIa/b after U2 snRNP deproteinization (Fig. 2). Similarly, Lührmann and colleagues observed increased reactivity of nucleotides 62-67 after deproteinization of U2 snRNP[44], which is incompatible with stem loop IIa but fully in agreement with the bulge in our proposed structure of naked U2 snRNA (Fig. 2). The authors further noticed reduced reactivity indicating stronger base-pairing for nucleotides upstream and downstream of the Sm site[44], which is consistent with the formation of NSS predicted by our in silico modeling and structural probing (Figs. 1 and 2).

While U2 snRNA adopts the alternative folding even without the extra 3′ end sequence, the 3′ end has effect on maturation of other pre-snRNAs. The SHAPE-MaP studies of in vitro transcribed U4 pre-snRNA was consistent with our in silico modeling (Figs. 1 and 2) and as well as with RNase probing by Myslinski et al.[77]. The partial reactivity of nucleotides 124 (U) and 125 (C) that are a part of the putative NSS indicates that this region is unstable and might spontaneously open (Fig. 2). However, deletion of Gemin3 prevented Cajal body localization of microinjected U4 pre-snRNA, which suggests that under in vivo conditions, NSS is stably formed and blocks Sm ring assembly. Identically to U2 snRNA, the mutational relaxation of U4 NSS removed requirements for Gemin3 (Fig. 5, U4noNSS) while strengthening NSS inhibited Cajal body localization (Fig. 5, U4stNSS), which supports the hypothesis that the structural context of NSS is important for proper snRNA biogenesis.

However, the situation with U1 pre-snRNA seems more complex. To model U1 pre-snRNA structure, we had to apply additional constraint mimicking binding of SNRNP70 protein, which indicates that U1 pre-snRNA folding is not spontaneous and requires additional factors to chaperon the correct U1 pre-snRNA structure. Our in silico predictions showed that U1 pre-snRNA folds depends on the extra 3′ sequence (compare Fig. 1 and S16). These various U1 pre-snRNA 3′ extensions influence U1 snRNP biogenesis. Microinjected U1-1 pre-sRNAs did not reach Cajal bodies indicating that it did not acquire the Sm ring (Fig. S16), while an alternative 3′ sequence supported Sm ring formation as shown by Cajal body accumulation of microinjected U1-P26 pre-snRNAs (Fig. 6). Currently, we do not have a clear explanation why various 3′ extensions affect differently the U1 snRNA fate but the correct folding could be one of the main reasons. The structural flexibility of U1 snRNA might reflect alternative U1 snRNA functions in splicing, transcription and telescripting[78]. Together, these data clearly show the impact of 3′ end sequence on snRNA and snRNP biogenesis and specifically on U1 snRNA.

The folding of primary transcripts includes extensive base-pairing around the Sm motif and often includes a few nucleotides from the Sm site. The presence of the NSS therefore represents a substantial barrier for Sm core formation mainly due to a steric hindrance making the Sm site inaccessible for SMN and Sm proteins. Based on our results we propose a structural rearrangement pathway that unwinds NSS and releases Sm motif nucleotides prior to Sm ring formation. This step could be the initial starting point for snRNP formation. The Sm motif and its vicinity represent sequence segments with highest occurrence

of uridines in all snRNAs. This makes this region highly negatively charged with respect to the rest of snRNA sequences. In addition, the presence of uridines makes structures encompassing Sm sites the least thermodynamically stable structural segments within the predicted pre-snRNA (Fig. S11). The relative thermodynamic instability most likely causes this part of the snRNA to be most susceptible for rearrangements of its secondary structure and NSS destabilization might be additional important role of the uridines besides of binding the Sm core. However, the need of folding constraints in our in silico modeling of the pre-snRNA restructuring pathway indicated that NSS unwinding is not an intrinsic property of pre-snRNA structures themselves, but requires extrinsic factor(s) to initiate the rearrangement. Based on these findings we speculate that the SMN complex, which interacts with pre-snRNA exported from the nucleus, opens the least thermodynamically stable structural segment (NSS) and induces larger structural changes in the pre-snRNA molecule.

Gemin3 is an essential protein in *Caenorhabditis elegans*, *Drosophila melanogaster,* and *Mus musculus*[28,79–81], but its exact cellular function was unclear. We identified an ATP-dependent NSS opening activity in the SMN complex, which was reduced upon depletion of Gemin3 (Fig. 7). This result together with our Gemin3 knockdown experiments point to Gemin3 as the factor responsible for NSS relaxation. Gemin3 belongs to a large family of DEAD-box helicases that function mainly as chaperons during assembly of various ribonucleoprotein[28,82]. DEAD-box helicases are often non-processive and consume ATP to separate only a few internal nucleotides to permit protein binding or further RNA rearrangements during biogenesis of ribonucleoprotein particles[83]. This is exactly the function proposed here for Gemin3, acting on internal duplexes in NSS and opening them up to allow Sm protein binding. When the duplex is strengthened as in stNSS constructs, the helicase is not capable to separate the strands and the snRNP biogenesis is inhibited (Figs. 3 and 6). Consistent with this suggestion, ATP-dependent RNA unwinding activity has been proposed for Gemin3, and cytoplasmic extracts depleted of Gemin3 exhibit reduced Sm core assembly activity[12,15,29,84] (Fig. 6). The function of Gemin3 in NSS unwinding is also supported by evolutionary data. Our in silico modeling indicates that NSS and its relaxation is conserved in Metazoa (Figs. 1 and 4), which coincides with the appearance of Gemin3 in evolution[28,85].

It should be noted that Sm core formation using a reconstituted SMN complex is not dependent on Gemin3 and ATP[17,23–25]. The likely explanation is that the snRNA substrates prepared by in vitro transcription might adopt alternative structures and the reaction solution contains a mixture of differently folded snRNAs (see also different structural folds for U1 snRNA in Fig. 1). A fraction of molecules hence spontaneously adopts the structure compatible with Sm protein binding. Furthermore, it has been already proposed that the SMN complex is a Brownian machine that couples spontaneous conformational changes driven by thermal energy to the directed delivery of Sm proteins onto snRNA, at least in vitro[17]. Finally, most in vitro assays did not utilize 3′ extended precursors, which at least in case of U1 significantly affects the NSS structure. In cells and cell extracts, pre-snRNAs are either fixed in the compact structure with NSS or their transition between different structures can be blocked by auxiliary protein factors that bind to snRNA. Under those conditions, Gemin3 becomes essential for snRNP biogenesis as the factor that actively initiates NSS opening[15,84]. Such a mechanism would also explain the ATP requirement for Sm ring assembly in cell extracts. It was shown that SMN complex-driven assembly of Sm protein on U1, U4, and U5 snRNAs is Gemin3-independent but depletion of Gemins 3-5 inhibited Sm protein association with U2 snRNA under the same conditions[17]. This suggests that different snRNAs might not have the same requirements for snRNP biogenesis factors.

In summary, we propose a previously undescribed compact architecture for human pre-snRNAs that needs to be disrupted prior to

Sm protein loading. The compacted structures might protect the naked pre-snRNAs and prevent their degradation. Indeed, removal of the Sm site or shortening/misprocessing of the 3′ end leads to uridylation and rapid degradation of snRNAs[86–90]. Compacted pre-snRNA structures might also serve as a checkpoint allowing the SMN complex to discriminate a correct pre-snRNA substrate from a random RNA molecule containing a stretch of uridines. Finally, we suggest that Gemin3 (together with Gemin4) is the factor that relaxes the compact pre-snRNA substructure involving the Sm motif and allows assembly of the Sm ring. The evolutionary conservation of Gemin3, predicted structures and the restructuring folding pathway suggests a functional relevance of pre-snRNA structures described here in snRNP biogenesis.

## Methods
### Computational procedures
**Prediction of suboptimal secondary structures.** The computational procedures in the presented work were based on predicted suboptimal secondary RNA structures. The structures were predicted using both unconstrained and constrained prediction of suboptimal secondary RNA structures, as specified in the "Results" section. The unconstrained prediction was carried out using UNAfold[58] with parameters W, P, and N set to 1, 1000, and 20, respectively. The constraint prediction was carried out using RNAsubopt[59] with default parameters. Similarity of suboptimal structures was computed using RNAdistance[59] in the form of tree edit distances. Following theoretical background was implemented using Matlab scripts described in Supplementary software 1.

**Identification of best representative suboptimal structures (step 9 in Fig. 1a).** We had $n$ species for one pre-snRNA, for which pre-snRNA sequences were available and which were enough evolutionarily distant from each other at the same time as explained in the "Results" section. $n$ for the individual analyzed pre-snRNAs were shown, explained and discussed in the main text of the manuscript. For each species we had $m_i$, $i = 1,...,n$, sequence variants of a single pre-snRNA, given by data available in databases. Then, we had to choose the variant for each species whose secondary structure was best representative for the structure of the corresponding pre-snRNA.

To that end, for the $j$th sequence variant, with $j = 1,...,m_i$, $r_j$ suboptimal structures were predicted, where $r_j \epsilon <1, 20>$, resulting in the total $\sum_{j=1}^{m_i} r_j$ suboptimal structures for the $i$th species, with $i = 1,...,n$. Among the suboptimal structures of each of the species, a best representative structure was found as a structure with the highest average similarity of its best matches to suboptimal structures of all variants given by the minimal average tree edit distance:

$$\min\left(\frac{\sum_{l=1}^{\sum_{p=1}^{mi} rp} \min\left(d_{l,k=1,...,r_j}\right)}{n}\right),\qquad(1)$$

where $j = 1,...,m_i$, $i = 1,...,n$, and where $d_{l,k}$ was a tree edit distance of the $l$th suboptimal structure of $j$th sequence variant to the $k$th suboptimal structure of $i$th species.

The suboptimal structures identified using the term (1) were considered the best representative structures. They were identified for all species of all pre-snRNAs, one best representative structure for one species. Note that the best representative structure belonged to one of the sequence variants of a species.

**Identification of homologous structures (step 11 in Fig. 1a).** Now we had best representative structures for species of analyzed pre-snRNAs. The trouble was that these structures did not need be similar to each other from principal reasons related to the RNA secondary structure

prediction, i.e., they had not have interspecies similarity although they were structures of single RNAs from homologous species. We therefore had to identify homologous structures.

We therefore, for each best representative structure computed as explained in the previous section, identified most similar suboptimal structures among suboptimal structures of other species. To that end, to the $i$th best representative structure, where $i = 1,...,n$, the most similar structure in the sets of suboptimal structures of the other $n-1$ species was identified with the minimal tree edit distance to the $i$th best representative structure given as $\min\left(d_{l,k=1,...,\sum_{j=1}^{m_i} r_j}\right)$, where $l = 1,...,n, i = 1,...,n$, and $l \neq i$, and $d_{l,k}$ was a tree edit distance between the $l$th template and $k$th suboptimal structure within the $i$th set. The condition $l \neq i$ prevents searching for the best matching suboptimal structure to the template of the same species, which is the template itself.

This way, to each best representative structure of a single species, one similar suboptimal structure from other species was identified. The result were $n$ sets, one set for one species, each set containing suboptimal structures from the other evaluated species similar to the best representative structure of the set, and therefore to each other. The sets were computed for all species of all pre-snRNAs. See Supplementary software 1 for details.

**Identification of sets with most mutually related structures (step 13 in Fig. 1a).** Among the sets of homologous structures computed as explained in the previous section, the sets with most mutually similar structures were identified for every analyzed pre-snRNA. They had minimal mutual tree edit distance, given as $\min\left(\sum d_{j=1,...,n,k=j+1,...,n}\right)$, where $d_{j,k}$ was a tree edit distance between $j$th and $k$th suboptimal structures within the $i$th set, $i = 1,...,n$.

These sets, one for one analyzed pre-snRNA, contained each $n$ suboptimal structures for $n$ species, similar to each other, thus representing secondary structures of homologous species of a single RNA. These structures were considered as best structural representatives of the analyzed pre-snRNAs in the evaluated species as close as theoretically possible to the native pre-snRNA structures. They were termed as primary folding transcripts and were used as model structures further in the presented study for the modeling structure rearrangement. See Supplementary software 1 for details.

**Computational identification of most mutually similar structures of folding intermediates and final structures (Fig. 4a).** We had to identify most mutually similar structures for folding intermediates (obtained in steps 1–5 in Fig. 4a) and final structures (obtained in steps 5–9 in Fig. 4a).

We had one primary folding transcript for each of pre-snRNA species for all analyzed pre-snRNAs, obtained according to the flow chart in Fig. 1a. For each of the primary folding transcripts, a number of suboptimal structures was predicted using constrained prediction, with the constraint simulating molecular interaction with Gemin3 protein. How many structures were predicted for each primary folding transcript depended on the prediction algorithm, which was RNAsubopt, but no more than the first 20 suboptimal structures were used to keep the task computable on a cca 80 Intel core cluster. The constraints used in the prediction, designed as described in the "Results" section, are shown in FASTA format in Supplementary file S1 for primary folding transcripts of individual pre-snRNAs.

Then, for $n$ folding intermediates for a single pre-snRNA, each for one evaluated species, we predicted $o_i$, $i = 1,...,n$, suboptimal structures, where individual numbers $o_i$ were given by the prediction algorithm, and $o_j \in \langle 1, 20 \rangle$, resulting in $n$ sets of suboptimal structures, one set for one species. Altogether, we obtained $\sum_{i=1}^{n} o_i$ suboptimal structures, where $i = 1,...,n$.

Next, among those $\sum_{i=1}^{n} o_i$ suboptimal structures, we identified most similar suboptimal structures to each of the folding intermediates from the species other than the species of the folding intermediates, based on their mutual minimal tree edit distances, i.e.,

$$\min\left(d_{j=1,...,\sum_{i=1}^{n} o_i, k=1,...,\sum_{l=1}^{n} o_l,}\right), \quad i \neq l,$$ i.e., using the folding intermediates as structural templates.

This way we obtained $n$ sets, one set for one template, each with $n$ suboptimal structures belonging each to one of the evaluated species. Now the task was to compute which of the templates was the best representative for folding intermediates secondary structure and we did it based on mutual similar of the suboptimal structures within the sets. Therefore we computed the set with the highest mutual similarity of its suboptimal structures, i.e., with the minimal mutual tree edit distance, as having $\min(\sum d_{j=1,...n,k=j+1,...,n})$, where $d_{j,k}$ was a tree edit distance between $j$th and $k$th suboptimal structures within the $i$th set, $i = 1,...,n$. This set was supposed to contain the most representative structures of the folding intermediates, as it contained mutually similar secondary structures for the largest number of evaluated species of all the sets. See Supplementary software 1 for details.

The above described procedure was also applied to final structures. The constraints for prediction of final structures, designed as described in the "Results" section, are shown in FASTA format in Supplementary Data 2 for individual pre-snRNAs.

## Experimental procedures

**Cell culture.** HeLa and HeLa S3 cells were cultured in Dulbecco's modified Eagle's medium containing 4.5 g glucose/l (Sigma) supplemented with 10% (HeLa) or 5% (HeLa S3) fetal bovine serum and 1% penicillin and streptomycin (Gibco).

**Plasmids.** The mutants of U2 (U2weakNSS, U2noNSS and U2stNSS) were created by site-directed mutagenesis using specific primers listed in Supplementary Data 3 and confirmed by sequencing. The U2-MS2 RNA construct, which includes the promoter sequence, was described previously[30]. The U2weakNSS-MS2 construct was prepared by site-directed mutagenesis using specific primers listed in Supplementary Data 3. U1-1 (GRCh38/hg38:chr1:16,840,617–16,840,779), U1-26P (GRCh38/hg38:chr14:35,025,383–35,025,595), U4-1 (GRCh38/hg38:chr12:120,730,865–120,731,040), and U5F-1 (GRCh38/hg38:chr1:44,721,744–44,721,901) pre-snRNAs were designed and synthesized by GeneArt service (Thermo Fisher Scientific) including variants containing mutations strengthening (stNSS) and relaxing NSS (noNSS).

**Establishment of the DDX20-EGFP-FKBP12[F36V] cell line.** Two targeting guides for DDX20/Gemin3 with MIT specificity score 70 and 73, respectively, were designed using http://crispor.tefor.net/ (see Supplementary Data 3). Oligonucleotides sgRNA70-pX330_F + sgRNA70-pX330_R, and sgRNA73-pX330_F + sgRNA73-pX330_R were annealed and inserted into the pX330-U6-Chimeric_BB-CBh-hSpCas9 (Addgene, #42230) plasmid[91] using BbsI restriction site to generate pX330-DDX20-sg70 and pX330-DDX20-sg73 plasmids containing the coding sequences of DDX20-specific sgRNA, and human codon-optimized S. pyogenes Cas9. To develop the pcDNA5/FRT-DDX20-mAID-EGFP-FKBP12[F36V] vector, the human DDX20 (Gemin3) coding sequence was amplified from cDNA using primers DDX20-KpnI F and DDX20-KpnI R and inserted into pcDNA5/FRT miniAID-EGFP (Addgene, #101713)[92] using the KpnI restriction site, which was introduced into the plasmid. Next, the first 1060 nucleotides of the DDX20 3'UTR were amplified using DDX20_RA_NotI_F and DDX20_RA_PstI_R primers and inserted into pcDNA5/FRT-DDX20-mAID-EGFP vector using NotI/PstI restriction sites. Finally, FKBP12[F36V] was amplified from pLEX_305-C-dTAG

(Addgene, #91798) plasmid[71] using G3-dTAG_GA_F and G3-dTAG_GA_R primers and assembled into PCR-amplified pcDNA5/FRT-DDX20-mAID-EGFP-3'UTR plasmid using primers Gem3-vect_GA_F and Gem3-vect_GA_R and the NEBuilder HiFi DNA Assembly Cloning Kit (New England Biolabs) according to the manufacturer's instructions, which created the DDX20-mAID-EGFP-FKBP12[F36V] vector. In parallel, we created the pRR-Puro-DDX20 vector. Primers DDX20_tar_pRR_F and DDX20_tar_pRR_R containing the DDX20 target sequence recognized by the sgRNAs were annealed and inserted into SacI/AatII double-digested pMB1610_pRR-Puro (Addgene, #65853) plasmid[93].

HeLa cells were co-transfected with pX330-DDX20-sg70, pX330-DDX20-sg73, pRR-Puro-DDX20, and DDX20-mAID-EGFP-FKBP12[F36V] plasmids using Lipofectamine 3000 (Thermo Fisher Scientific) according to the manufacturer's instructions. Subsequently, 24 h post-transfection, cells were selected in fresh media containing 1 μg/mL puromycin for 72 h and EGFP-positive single cell clones were selected and expanded. Genomic DNA was isolated using High Pure PCR Template Preparation Kit according to the manufacturer's instructions. To genotype the cells we isolated genomic DNA and confirm the knock-in by Q5 High-Fidelity DNA Polymerase (New England Biolabs), and primers DDX20-gentotype F and R. Amplified DNA was purified by gel extraction using Zymoclean Gel DNA Recovery Kit (Zymo Research) and sequenced. One obtained homozygous clone was also confirmed by western blotting using α-Gemin3 mouse monoclonal antibody (clone 12H12; Santa Cruz Biotechnology, catalog# sc-57007; Fig. S17a). For Gemin3-depletion, the HeLa DDX20-mAID-EGFP-FKBP12[F36V] cells were plated and after 24 h the cells were induced with 0.5 μM dTAG13 (Tocris) for 72 h (Fig. S17b, c).

**Antibodies.** For indirect immunostaining, we used mouse monoclonal anti-coilin (5P10) antibody (dilution 1:1000), kindly provided by M. Carmo-Fonseca (Institute of Molecular Medicine, Lisboa). Anti-Gemin3 (mouse monoclonal, clone 12H12, Abcam, catalog# ab10305 or Santa Cruz Biotechnology, catalog# sc-57007, dilution 1:400), anti-Gemin4 (mouse monoclonal, clone 3E1, Sigma, catalog# WH0050628M1-100UG, dilution 1:500) and anti-Gemin5 (mouse monoclonal, clone 10G11, SantaCruz Biotechnology, catalog# sc136200, dilution 1:500) were used for western blotting. Mouse monoclonal anti-SMN antibody (clone 7B10 [94], ImmunoGlobe, catalog# 0176-01) was used for the SMN complex purification. This antibody was prepared from original hybridoma cell line by Archana Prusty (Department of Biochemistry, Theodor Boveri Institute, University of Würzburg). Secondary anti-mouse antibodies conjugated with Alexa-647 (Thermo Fisher Scientific, Cat No. A21236) were used for immunofluorescence and peroxidase-conjugated anti-mouse IgG (Jackson ImmunoResearch Laboratories, Cat No. 115-035-003), peroxidase-conjugated anti-rabbit IgG (Jackson ImmunoResearch Laboratories, Cat No. 111-035-003). For immunoprecipitation, we used anti-Sm Y12 antibody produced from a hybridoma cell line (a gift from Karla Neugebauer, Yale University, New Haven, USA) at the Antibody Facility (Institute of Molecular Genetics of the Czech Academy of Sciences).

**RNAi.** The siRNAs (Invitrogen) used in this study against Gemin3 (GCAUACAUAUGGUAUAGCAtt, s22143, Ambion), Gemin4 (GGCA-CUGGCAGAAUUAACAtt, custom design, Ambion) and Gemin5 (GAAAUACGGCAACACGAAAtt, s24773, Ambion) were transfected using Oligofectamine (Invitrogen) according to the manufacture's protocol to a final concentration 20 nM. Cells were microinjected 72 h (siRNA Gemin3 and Gemin4) or 48 h (siRNA Gemin5) after transfection. The negative control No. 5 siRNA from Invitrogen was used as a negative control. The efficiency of Gemin3, Gemin4, and Gemin5 knockdowns is evaluated at Fig. S18a.

**In vitro transcription.** All DNA templates for in vitro transcription were prepared by PCR using Phusion polymerase (Biolab) using primers

listed in Supplementary Data 3. Fluorescently or radioactively labeled RNAs were prepared as described previously[30] by in vitro transcription using a MEGAshortscript kit (Thermofisher) containing UTP-Alexa 488 (Invitrogen) or radioactive αUTP (Hartmann Analytic) and trimethylated cap analog (m3 2,2,7G(5)ppp(5)G (Jena Bioscience)). After synthesis, RNA was isolated by phenol/chloroform extraction, precipitated, and dissolved in nuclease-free water. RNA was diluted in a solution containing dextran-TRITC 70-kDa (Sigma-Aldrich) to final concentration 200 ng/l.

**Microinjection.** HeLa cells were grown on glass coverslips and RNA was microinjected using InjectMan coupled with FemtoJet (Eppendorf) as described previously[30,95]. For microinjection of denatured U2WT snRNA, RNA was incubated at 98 °C for 5 min and immediately microinjected into the HeLa cells. After 1 h incubation period, cells were rinsed twice with PBS and fixed for 20 min at room temperature in 4% PFA/PIPES (freshly prepared).

**Indirect immunofluorescence and image acquisition.** HeLa cells grown on coverslips were fixed, labeled, and images were acquired using the DeltaVision microscopic system (Applied Precision) coupled to Olympus IX70 as described previously[96]. Stacks of 20 z-sections with 200 nm z steps were collected per sample and subjected to mathematical deconvolution using SoftWorx software. Maxima projections of deconvoluted pictures were generated by SoftWorx and are presented. ImageJ was used to determine the fluorescence intensity in Cajal bodies and nucleoplasm of microinjected cells.

**RNA isolation (in vitro, in vivo, and ex vivo) and SHAPE probing.** Details of RNA sample preparation and data processing have been described in detail[97]. For the in vitro experiment, U2WT and U4 pre-snRNA were transcribed by T7 polymerase P266L variant[98] from pcDNA3 or p-MA-T plasmids, respectively, with T7 promoter followed by DNase I (30 min at 37 °C) and Proteinase K (30 min at 37 °C) treatments. U2 snRNA was purified on 30 kDa Amicon columns. Then, RNA was folded for 30 min at 37 °C in 57 mM $MgCl_2$ followed by labeling with 100 mM NAI at 37 °C for 10 min. DMSO was used as a negative control. U4 pre-snRNA was purified on Superdex 200 Increase 10/300GL colomn pre-equilibrated with filtration buffer (50 mM K-HEPES pH 7.4, 150 mM KCl, 0.1 mM EDTA). Peak fraction was then diluted to 100 ng/μL and folded for 30 min at 37 °C in 60 mM $MgCl_2$ followed by labeling with 100 mM NAI at 37 °C for 10 min. DMSO was used as a negative control. For the ex vivo experiment, RNA was isolated from HeLa cells. Cells were washed 3x with 1xPBS, dissociated by Trypsin-EDTA solution (Sigma) and collected by centrifugation at $500 \times g$ and 4 °C for 5 min. Cells were resuspended in 5 ml lysis buffer (40 mM Tris-HCl pH 8.0, 25 mM NaCl, 6 mM $MgCl_2$, 1 mM $CaCl_2$, 256 mM sucrose, 1000 U/ml SUPERase-IN RNAse inhibitor, 0.5% Triton X-100, 450 U/ml DNAse I), rotated 5 min at 4 °C and pelleted at $2250 \times g$ for 2 min at 4 °C. The pellet was resuspended in proteinase K buffer (100 mM Tris-HCl pH 7.5, 200 mM NaCl, 2 mM EDTA, 1% SDS, 500 μg/ml Protease K) and incubated at room temperature for 45 min. After incubation, pre-equilibrated phenol/chloroform/isoamyl alcohol buffered by folding buffer (110 mM HEPES pH 8.0, 110 mM KCl, 11 mM $MgCl_2$) was added and samples were centrifuged at $12,000 \times g$ for 15 min at 4 °C. RNA was cleaned on a PD-10 column according to the manufacturer's instructions. 100 mM NAI or DMSO was added to elute RNA and incubated for 10 min at 37 °C. For the in vivo experiment, we started with $10 \times 10^6$ HeLa cells. Cells were 1x washed by 1xPBS and collected by centrifugation for 5 min at $10,000 \times g$ and 4 °C. The pellet was resuspended in 500 μl 1xPBS and split into two tubes. NAI or DMSO to final concentration 200 mM was added and cells were incubated for 10 min at 37 °C. RNA was isolated using Trizol (Sigma) and 200 μl chloroform and precipitated with ethanol at −20 °C overnight. All prepared RNA samples (in vitro, ex vivo, in vivo) were used for reverse transcription

with the gene-specific primer 5′-CGTTCCTGGAGGTACTGCAA for U2 snRNA and 5′-AAAAATTCAGTCTCCG for U4 pre-snRNA. We used SHAPE MaP buffer (50 mM Tris·HCl pH 8.0, 75 mM KCl, 10 mM DTT, 0.5 mM dNTP, 6 mM MnCl$_2$) and SuperScript II (Invitrogen).

**Preparation of sequencing libraries.** Amplicons for snRNAs were generated using gene-specific forward and reverse primers. Importantly, the primers include Nextera adapters required for downstream library construction. All gene-specific and adapter sequences used are detailed in Supplementary Data 3. Gene-specific amplicons were generated using these primers, 5 μL of purified cDNA, and NEBNext Ultra II Q5 MasterMix (Cat. No. M0544L). PCR reaction products were cleaned using Monarch PCR&DNA Clean-up Kits (NEB, Cat. No. T1030S) and a binding buffer:sample ratio of 5:1. Remaining Illumina adapter sequences were added using the PCR MasterMix and index primers provided in the NexteraXT DNA Library Preparation Kit (Illumina) according to the manufacturer's protocol, but using 1/5th the suggested reaction volumes. Libraries were quantified using Qubit (Invitrogen) and BioAnalyzer (Agilent). Amplicons were sequenced on a NextSeq 500/550 (U2) and NextSeq 2000 (U4) platforms using a 150 cycle mid-output kit.

**SHAPE-MaP data analysis.** All sequencing data was analyzed using the ShapeMapper 2 analysis pipeline[99]. The '−amplicon' and '−primers' flags were used, along with sequences of gene-specific handles PCR primers, to ensure primer binding sites are excluded from reactivity calculations. Default read-depth thresholds of 5000x were used. Analysis of statistically significant reactivity differences between ex vivo and in vivo-determined SHAPE reactivities was performed using the DeltaSHAPE automated analysis tool (https://github.com/Weeks-UNC/deltaSHAPE) and default settings[67,100]. Regions where 3 of 5 nucleotides passed both criteria Z factor >0 and standard score ≥1 were considered significant.

**SMN complex purification.** HeLa S3 cells were pelleted at 1000 × $g$ for 5 min, washed once with 1xPBS and pelleted again before snap freezing. The pellets were thawed on ice and resuspended in lysis buffer (1xPBS, 0.01% NP-40, 2.5 mM MgCl$_2$, 0.8 U/μL murine RNase inhibitor, 1:1000 protease inhibitors). After incubation on ice for 10 min, the cells were lysed using a dounce homogenizer with tight pestle. After centrifugation at 10,000 × $g$ for 10 min, required volume of supernatant was incubated with ProteinG-Sepharose beads (GE Healthcare) covalently coupled to anti-SMN antibody (monoclonal 7B10[94], ImmunoGlobe, 0176-01), for 3 h at 4 °C on a head-over-tail rotor for immunoprecipitation. After three washes using wash buffer (1xPBS, 0.01% NP-40, 2.5 mM MgCl$_2$) and twice with storage buffer (1xPBS, 2.5 mM MgCl$_2$), the beads were resuspended in equal volume of storage buffer to a 50% slurry and analyzed by SDS-PAGE followed by Coomassie staining or western blotting (Figs. S17d and S18b) or used for assays.

**In vitro Sm ring assembly with Sm proteins.** HeLa cells were cultivated on 15 cm Petri dish and treated by Gemin3 or Negative control siRNAs for 72 h prior harvesting. The cytoplasmic extract was prepared using the NE-PER™ Nuclear and Cytoplasmic Extraction Reagents (Thermofisher) following the manufacture protocol. In vitro transcribed U2 snRNA WT and mutants were in vitro transcribed using the MEGAshortscrip T7 Kit (Thermofisher) and labeled by radioactive [α-$^{32}$P]UTP. In vitro transcribed RNAs were then heat-denatured for 90 s at 80 °C and placed on ice. RNAs were mixed with cytoplasmic extract and incubated at 37 °C for 1 h. Then, snRNPs were immunoprecipitated using the anti-Sm Y12 antibody, RNA was extracted by phenol/chloroform, precipitated and resolved in polyacrylamide gel containing 7 M urea and detected using the imaging phosphor screen

(GE Healthcare) for 12 h and developed by Typhoon 9000 (GE Healthcare).

**In vitro Sm ring assembly with purified SMN complex.** Two μg of in vitro transcribed snRNA were incubated with 25 μl of the purified SMN complex with 4.5 mM ATP, 3 mM MgCl$_2$ for 30 min at 37 °C. After the incubation, the samples were briefly pelleted by centrifugation and the supernatant was used for immunoprecipitation. Immunoprecipitation was performed as previously described[88] using the mouse anti-Sm Y12 antibody. RNA was extracted using phenol/chloroform, precipitated and resolved in polyacrylamide gel containing 7 M urea and radioactivity detected by imaging phosphor screen (GE Healthcare) and developed by Typhoon 9000 (GE Healthcare).

**Molecular beacon assay.** Molecular beacon was synthesized by Sigma-Aldrich with a quencher on the 5′ end and Texas red on the 3′ end. Alternatively, the molecular beacon was assembled from two RNA primers when the quencher was attached to 5′ end of one primer and Texas red to 3′ end of the second primer (Fig. 7c). Before the incubation with the SMN complex, 10 mM molecular beacon was denatured at 98 °C for 5 min and slowly cooled down for 3 h to room temperature. Then, the beacon was incubated with a purified SMN complex and ATP (4.5 mM). Alternatively, the SMN complex was pre-incubated with ATP-γS (10 mM) for 1 h at 37 °C. Fluorescence was measured in a 96-well black assay plate at 594 nm using a Mithras LB 940 reader (Berthold Technologies).

**Statistics and reproducibility.** Gels and micrographs presented at Figs. 3c, 5c and 7a represent experiments that were repeated three times with similar results.

### Reporting summary

Further information on research design is available in the Nature Portfolio Reporting Summary linked to this article.

## Data availability

The data supporting the findings of this study are available from the corresponding authors upon reasonable request. The RNA-seq data used for SHAPE-MaP are accessible at ArrayExpress (www.ebi.ac.uk/biostudies/arrayexpress) using accession code E-MTAB-13248. Source data for the figures and supplementary figures are provided as a Source data file. Source data are provided with this paper.

## Code availability

The software code used to predict pre-snRNA structures and modeling of structural rearrangements Supplementary software 1.

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

## Acknowledgements

We thank Maria Carmo-Fonseca, Ina Poser, and Karla Neugebauer for providing us with reagents and Sarka Takacova for proofreading of the manuscript. This work was supported by the Czech Science Foundation (21-04132S to D.S.), ELIXIR CZ research infrastructure (MEYS Grant No: LM2023055) including access to computing and storage facilities (J.P.), DFG-grants (Fi573/25-1 to U.F.), and the institutional funding (RVO68378050 and RVO68378050-KAV-NPU1. N.R. was supported by the International Visegrad fund scholarship #52210798, N.H. was supported by the NIH training grant T32AI055403. A.M.P. is an HHMI Investigator. The microscopy images were acquired at the Light Microscopy Core Facility, Institute of Molecular Genetics in Prague, Czech Republic supported by MEYS (LM2015062, CZ.02.1.01/0.0/0.0/16_013/0001775) and OPPK (CZ.2.16/3.1.00/21547).

## Author contributions

J.P. performed all in silico modeling shown in Figs. 1 and 4 and Supplementary Figs. 1–15. A.R. performed all microinjections and experiments with Sm ring assembly shown in Figs. 3, 5, 6, and 7. N.H. and H.W. together with A.R. and N.R. performed SHAPE-MaP (Fig. 2). A.P. prepared the purified SMN complex (Figs. S18 an d S18) utilized for in vitro experiments (Figs. 3 and 7). M.S. prepared and characterized the Gemin3-degron cell line that was used to isolate the SMN complex lacking Gemin3 (Figs. 7 and S17). J.P., A.M.P., U.F. and D.S. conceived the project and wrote the manuscript.

## Competing interests

The authors declare no competing interests.
