## [Peer Review File · Nature Communications]

The SMN complex drives structural changes in human snRNAs to enable snRNP assemblyReviewers' Comments:

Reviewer #1:

Remarks to the Author:

Review of Panek et al. Nat Comms ms. # 21-22591-T

The authors use a nice combination of bioinformatics, molecular cell biology and RNomics to model structural rearrangements that may take place during the biogenesis of spliceosomal Sm-core snRNPs. If proven true, the identification of a paired NSS (Near Sm-site Stem) in precursor snRNAs provides a mechanistic basis for how the SMN complex uncovers the Sm binding site during core assembly. This represents an important finding. Furthermore, if Gemin3 is indeed important for driving these rearrangements that would also provide important detail in bolstering the model.

Respectfully signed, Greg Matera

Major comments:

Overall, the manuscript seems a bit thin on data – or perhaps a better way to say is that it feels slightly underdone. Particularly in two key areas, the authors leave gaps that need to be filled before the manuscript can be published. I think if they could flesh out these two main areas, the paper could be greatly improved. One area is bioinformatic and the other one will require additional wet-bench experiments.

1. First of all, with the exception of human pre-snRNAs, its not entirely clear if any of these metazoan 3'-extended precursors have been detected in vivo. We looked hard for them by northern blotting many years ago in flies and could not detect them. They could be short lived. Are the authors aware of any data along these lines in any of the other metazoans they analyze in Fig. 1? (This may be more of a minor point, but additional experimental evidence for the whole bioinformatic approach the authors take seems fundamental to me).

2. It is not clear from the manuscript or the supplemental figures and tables which species the authors consider as being a Protozoan. In looking over the supplementary table S1, it appears that *S.pombe* and *S.cerevisiae* have been lumped in with the protists. And on pg 9 the authors state something about "... in yeast and other protozoan U2 snRNAs." This is a huge mistake. And an opportunity (I hope!). A very large number of fungal genomes have been sequenced, and I believe the authors should re-do their folding analysis using fungi as a third group of organisms.

The opportunity comes from the fact that fungi are actually higher eukaryotes that are more closely related to metazoans than they are to protists. However, fungal genomes have lost several components of the SMN complex (in fact, *S.cerevisiae* only have Gem2 and do not even have an SMN gene -- and their U2 snRNA is truly wacky with a non-essential 1000nt insertion, so you might even exclude *cerevisiae* from the analysis proposed below).

Due to pressure to make their genomes smaller, *S.pombe* and other fungi have actually lost the Gemin3-4-5 subunit of the SMN complex from their genomes (along with MANY other genes), whereas it seems that protists never even had this subcomplex in the first place. Therefore, either fungi do not require an RNA helicase to assemble Sm-core RNPs, or some other helicase serves in place of Gemin3. But I digress.

The big question is this: Do the predicted fungal pre-snRNAs have a base-paired NSS or is the Sm-site of for this class of organisms mostly single stranded? If fungal pre-snRNAs do not form an NSS, then that might help explain why Gemin3 and Gemin4 have become dispensable. Some in vitro SHAPE analysis or other structure probing to confirm this prediction would be important here as well.

3. In Figures 5 and 6, the authors focus on RNAi experiments targeting Gemin3 and Gemin5 and they analyze the effects on nuclear import/Cajal body transport. Gemin3 knockdown has an effect but not Gemin5. The negative result for Gemin5 is fine, as this protein is really more of a separate subunit of its own. However, within the context of the SMN complex, Gemin3 has a dedicated binding partner in Gemin4.

Why have the authors not carried out RNAi on Gem4? Moreover, Gem4 is the only member of the SMN complex that actually contains a nuclear localization signal (NLS), and overexpression of this protein drives the SMN complex into the nucleus (see Meier et al. 2018). The authors should include Gem4 RNAi in the experiments shown in Figs 5 and 6. Note, there are published siRNAs for Gem4 that give good knockdown, so that's not an excuse (Shpargel 2005, PNAS).

Minor points

a. Top of pg 2 should read: "Spliceosomal snRNAs (except U6 and U6atac) are synthesized by RNA polymerase II (or 2)." That is, you left off the II.

b. Introduction: the authors don't provide enough motivation for why they focused so much attention on U2. Especially since there is some disagreement about the secondary structure of the U2 precursor. Most of the work on Sm core assembly in vitro has been done with U1 or U4. I assume this was because the assays involve microinjection and trafficking to Cajal bodies in HeLa cells? A number of years ago, we showed using digitonin permeabilized cells that fluorescently-labeled 12S U2 snRNP precursors accumulated in CBs after nuclear import, whereas mature U1 snRNPs did not (Ospina et al. 2005). This paper should be cited (perhaps along with refs 30-35?). Also, the sentence prior to that one needs some sort of citation (either a review or primary papers).

c. Pg. 4: "Various 3' end extensions reported for various snRNA genes..." Too vague.

d. In comparing figure S1 to S5, it would be helpful to show how the structure changes. Perhaps show an inset of the cartoon from Fig S1 inside panels of Fig S5?

e. The weak descriptor at the bottom of page 8 ("These data ... indicate that Gemin3 is a potential candidate for NSS opening.") summarizing what was actually shown contrasts with the strong statement in the Abstract regarding the role of Gemin3.

f. Middle page 5: "... we applied SHAPE-MaP analysis to compare..." This sentence needs reference to a Weeks lab paper (e.g. Smola et al. 2015?)

g. Fig. 1A, step 12. Should read "Each set contains..."

h. Pg. 10, middle: If you are going to specify that Gemin3 is an essential gene in the three individual species, then you should cite the primary literature. Otherwise just say it's an 'essential metazoan protein' and cite the review.

i. There may be a problem in the journal metadata, as the author list in the email I got from the journal is different than the author order that is shown in the PDF.

Reviewer #2:

Remarks to the Author:

The manuscript by Panek et al. reports the role of RNA helicase Gemin 3 in the biogenesis of snRNPs. Particularly, the authors found evolutionary conserved sequence motifs that would interfere the binding of Sm proteins and they show a potential role of Gemin 3 in destabilizing such sequence motifs for Sm core formation. Although the authors provide clear data to evidence the role of Gemin 3 in restructuring snRNAs, the study does not provide a conceptual advancement in the role of SMN complex in snRNP biogenesis. ATP- and Gemin 3-dependent snRNP assembly has been shown previously and the study only adds the finding of near Sm site sequences that need to be restructured. In addition, the study only used U2 snRNA as a model system to prove their hypothesis.

Major points)

1. The role of NSS in snRNAs is not clear in the entire study. Can authors show that NSS in all other snRNAs including minor spliceosomal snRNAs plays a similar role? Does NSS affect the binding of snRNAs to Gemin5?
2. The authors produced weak and stable NSS U2 mutants. It is surprising to find that the stable NSS is less efficient in forming snRNPs and localizing to Cajal bodies compared to weak (or relaxed) NSS U2. In this scenario, what does Gemin 3 do? RNA helicases in general are supposed to destabilize helical structures. Stable NSS is a perfect match of Sm sequences which are mostly A-U base pairs. It is difficult to imagine that an RNA helicase cannot unwind this strength of dsRNA.
3. Are NSSs in other snRNAs similar or are they different? Can authors compare the strength of NSS across snRNAs?
4. The correlation between NSS and Gemin3 is not clear in the current study.

Reviewer #3:

Remarks to the Author:

In this manuscript, Panek et al. analyze the secondary structure of pre-snRNAs and biogenesis of snRNPs, claiming that the structures of pre-snRNAs and rearrangements around the Sm binding sequence are evolutionally conserved in Metazoa. The authors also showed, for the first time, that Gemin3 helicase is involved in structural rearrangements of human U2 snRNAs. The results presented are interesting, but I have some important comments and suggestions that need to be addressed by the authors.

Major Comments:

1. There is not enough experimental data to support some conclusions from the abstract (e.g., Gemin3 function). The abstract has to be rewritten, or new experimental data must be added (please see the comments below).
2. I am concerned that all the experimental results focus entirely on human U2 snRNA. The authors apply in silico modeling to predict suboptimal secondary structure models for four pre-snRNA/sn-RNA from 9-11 species, but only one model (for one species) was experimentally confirmed. It is a little strange that the authors, familiar with a very sophisticated method of RNA structure mapping (SHAPE-Map), performed the SHAPE-Map experiment only for human U2 snRNA. Although the authors found that in vitro and ex vivo SHAPE reactivities match predicted U2 snRNA structure, there is growing evidence that RNA models predicted in silico without experimental constraints are often inconsistent with experimentally determined structures (Rice et al. 2014, doi: 10.1261/rna.043323.113; Busan et al. 2019, doi: 10.1021/acs.biochem.8b01218, figure 3; Deigan et al. 2009, doi: 10.1073/pnas.0806929106, tables 1 and 2; Siegfried et al. 2014, doi: 10.1038/nmeth.3029. Epub 2014 Jul 13, figure 3.

The manuscript would benefit from showing mapping and functional results for other snRNAs.

- 3. Moreover, the new function of Gemin3 helicase was also showed just for human U2 snRNA.

Minor Required Revision:

1. Sm – please provide the full name of this protein, e.g., in the introduction.
2. Page 2. "All spliceosomal snRNAs (except U6 and U6atac) are synthesized by RNA polymerase." Pls correct.
3. Page 3. Lines 18-20 It is accurate that a single MFE structure represents only one structure for a given sequence. However, the MFE algorithm also allows the prediction of multiple structures for one RNA sequence. Please clarify.
4. Page 5. Lines 34-39; I do not understand why SHAPE reactivities were mapped on in silico predicted structure or in vivo structure from another study. BTW Please, provide a source of in vivo model. Why did the authors not use the obtained SHAPE reactivities as pseudo-energy constraints for RNA structure modeling?
5. Page 6. Lines 1-4; It is pretty easy to test how Sm binding impacts SHAPE modifications using in vitro system similar to that described in the next chapter of the results section.
6. Discussion. Page 9. Lines 9-11; I think that stem-loops I, III, and IV are stable. Please correct me if I am wrong.
7. Page 14.; Could you explain why RNA was folded in 57 mM MgCl₂? It is a very high concentration for this type of experiment.
8. Page 22.; Legend of Figure 2. "... were mapped onto mature (ex vivo) or..." Probably in vivo is correct.

Reviewer #1

We thank the reviewer for positive feedback and constructive criticism that improved our work. Please, find below detailed answers to individual comments.

1. First of all, with the exception of human pre-snRNAs, its not entirely clear if any of these metazoan 3'-extended precursors have been detected in vivo. We looked hard for them by northern blotting many years ago in flies and could not detect them. They could be short lived. Are the authors aware of any data along these lines in any of the other metazoans they analyze in Fig. 1? (This may be more of a minor point, but additional experimental evidence for the whole bioinformatic approach the authors take seems fundamental to me).

The reviewer is correct that the data for pre-snRNAs from non-human species are missing. To overcome this issue, we estimated the length of 3' end extensions based on human results. We understand that this is only approximation. We therefore experimentally tested and validate the predicted structures for human snRNA by SHAPE-MaP. Because our main focus is put on human snRNAs, we did not experimentally validate other non-human snRNAs.

2. It is not clear from the manuscript or the supplemental figures and tables which species the authors consider as being a Protozoan. In looking over the supplementary table S1, it appears that *S.pombe* and *S.cerevisiae* have been lumped in with the protists. And on pg 9 the authors state something about "... in yeast and other protozoan U2 snRNAs." This is a huge mistake. And an opportunity (I hope!). A very large number of fungal genomes have been sequenced, and I believe the authors should re-do their folding analysis using fungi as a third group of organisms.

Big thanks for pointing this out! We completely recalculate the pre-snRNA structures and applied in silico modeling on 15 fungi species and 14 protazoan species (Table S1). These data indicate that the structural motif around the Sm site is conserved only in metazoans (Fig. 1d). We further discuss the evolutionary aspect of NSS and Gemin3 appearance in evolution in Discussion (p. 12, first paragraph).

The big question is this: Do the predicted fungal pre-snRNAs have a base-paired NSS or is the Sm-site of for this class of organisms mostly single stranded? If fungal pre-snRNAs do not form an NSS, then that might help explain why Gemin3 and Gemin4 have become dispensable. Some in vitro SHAPE analysis or other structure probing to confirm this prediction would be important here as well.

Our main aim in the first part of the manuscript was to determine structure of human pre-snRNAs. We applied an in-silico approach that relies on statistics (= identifying the most frequent structural folds) and comparison among species. The structural predictions of non-metazoan species were originally in the same group as metazoan, because we assumed that pre-snRNA structures are conserved among all eukaryotes. During the course of the work, we realized that metazoan pre-snRNAs stand out and have a conserved pre-snRNA structures while other species do not. We therefore sequestered metazoan and non-metazoan species and analyzed them separately. However, our main interest remained in human snRNAs and we therefore experimentally tested and validated the new predicted structures for human U2 snRNA and U4 pre-snRNAs (Fig. 2) and did not analyze other non-human samples.

3. In Figures 5 and 6, the authors focus on RNAi experiments targeting Gemin3 and Gemin5 and they analyze the effects on nuclear import/Cajal body transport. Gemin3 knockdown has an effect but not Gemin5. The negative result for Gemin5 is fine, as this protein is really more of a separate subunit of its own. However, within the context of the SMN complex, Gemin3 has a dedicated binding partner in Gemin4. Why have the authors not carried out RNAi on Gem4? Moreover, Gem4 is the only member of the SMN complex that actually contains a nuclear localization signal (NLS), and overexpression of this protein drives the SMN complex into the nucleus (see Meier et al. 2018). The authors should include

Gem4 RNAi in the experiments shown in Figs 5 and 6. Note, there are published siRNAs for Gem4 that give good knockdown, so that's not an excuse (Shpargel 2005, PNAS).

We have knockdown Gemin4 as suggested by the reviewer and observed a similar, albeit weaker, phenotype upon U2 snRNA microinjection. These data are included in Fig. 5a.

Minor points

a. Top of pg 2 should read: "Spliceosomal snRNAs (except U6 and U6atac) are synthesized by RNA polymerase II (or 2)." That is, you left off the II.

Corrected - thank you!

b. Introduction: the authors don't provide enough motivation for why they focused so much attention on U2. Especially since there is some disagreement about the secondary structure of the U2 precursor. Most of the work on Sm core assembly in vitro has been done with U1 or U4. I assume this was because the assays involve microinjection and trafficking to Cajal bodies in HeLa cells? A number of years ago, we showed using digitonin permeabilized cells that fluorescently-labeled 12S U2 snRNP precursors accumulated in CBs after nuclear import, whereas mature U1 snRNPs did not (Ospina et al. 2005). This paper should be cited (perhaps along with refs 30-35?). Also, the sentence prior to that one needs some sort of citation (either a review or primary papers).

We started our project with U2 snRNA because based on the prediction, the NSS formation does not depend on 3' end extension, which was then confirmed by SHAPE-MaP. In the revised version, we extended our experiments and included analysis of U1 and U4 pre-snRNAs. We showed that similarly to U2 snRNA, Gemin3 is important for accumulation of both snRNAs in Cajal bodies. In addition, U1 and U4 snRNA variants with weakened NSS did not require Gemin3, with is consistent with the proposed function of Gemin3 in relaxation of NSS structures in pre-snRNAs. These data are presented in new Fig. 6. We also tested localization of microinjected U5 pre-snRNA, but this RNA did not accumulate in Cajal bodies regardless of Gemin3 presence and we did not analyze U5 any further. We also added the references as suggested.

c. Pg. 4: "Various 3' end extensions reported for various snRNA genes..." Too vague.

In the revised version, we specify the range of pre-snRNA 3' end extensions.

d. In comparing figure S1 to S5, it would be helpful to show how the structure changes. Perhaps show an inset of the cartoon from Fig S1 inside panels of Fig S5?

A more detailed picture where a reader can directly compare different U1 structures is presented in Fig. 1 b and c. We believe that this is a better way to present the data because Fig. S5 is already busy and addition of an inset would make it even more filled.

e. The weak descriptor at the bottom of page 8 ("These data ... indicate that Gemin3 is a potential candidate for NSS opening.") summarizing what was actually shown contrasts with the strong statement in the Abstract regarding the role of Gemin3.

We added new experimental evidence that Gemin3 (together with Gemin4) is the factor that restructures the pre-snRNA template to allow Sm ring formation (Figs. 5-7). We therefore made the statement at the end of Results (p. 9, last paragraph of Results) stronger and in line with Abstract.

f. Middle page 5: "... we applied SHAPE-MaP analysis to compare..." This sentence needs reference to a Weeks lab paper (e.g. Smola et al. 2015?)

Done.

g. Fig. 1A, step 12. Should read "Each set contains..."
Corrected, thank you!

h. Pg. 10, middle: If you are going to specify that Gemin3 is an essential gene in the three individual species, then you should cite the primary literature. Otherwise just say it's an 'essential metazoan protein' and cite the review.
We kept the original statement and added references to primary literature.

i. There may be a problem in the journal metadata, as the author list in the email I got from the journal is different than the author order that is shown in the PDF.
Thank you for pointing this out but I was not able to modify the author order in the submission system. That's why the author order in metadata differs from the manuscript file. The authors order in the manuscript file is the correct one.

Reviewer #2

The manuscript by Panek et al. reports the role of RNA helicase Gemin 3 in the biogenesis of snRNPs. Particularly, the authors found evolutionary conserved sequence motifs that would interfere the binding of Sm proteins and they show a potential role of Gemin 3 in destabilizing such sequence motifs for Sm core formation. Although the authors provide clear data to evidence the role of Gemin 3 in restructuring snRNAs, the study does not provide a conceptual advancement in the role of SMN complex in snRNP biogenesis. ATP- and Gemin 3-dependent snRNP assembly has been shown previously and the study only adds the finding of near Sm site sequences that need to be restructured.

Gemin3 is an essential protein in several animal species but its specific function during snRNP biogenesis has not been uncovered. To address this reviewer's concern, we performed additional experiments to probe Gemin3 function. Namely, we mutated pre-snRNAs to weaken NSS and showed that these pre-snRNAs do not require Gemin3 for their biogenesis, in contrast to wild-type pre-snRNAs (Figs. 5 and 6). In addition, we isolated the SMN complex lacking Gemin3 (and Gemin4 and 5) and showed that depletion of Gemin3 reduces ability of the SMN complex to relax U2 NSS (Fig. 7). We believe that the new data together with previous results convince the reviewer that our manuscript represents conceptual advance in understanding of the SMN complex, and particularly Gemin3, function during snRNP biogenesis.

In addition, the study only used U2 snRNA as a model system to prove their hypothesis.

To address the reviewer's concern, we added structural analysis of U4 pre-snRNA by SHAPE-MaP (Fig. 2) and included analysis of U1 and U4 pre-snRNA (and their variants with weakened and strengthened NSS) upon Gemin3 knockdown (new Fig. 6). All these data support our main conclusions that Gemin3 is important for relaxation of compact pre-snRNA structures to allow Sm protein binding.

Major points

1. The role of NSS in snRNAs is not clear in the entire study. Can authors show that NSS in all other snRNAs including minor spliceosomal snRNAs plays a similar role?

The reviewer is absolutely correct that the role of NSS is not clear. We do not know, whether NSS has any particular role in snRNP biogenesis or whether its formation is just a by-product and undesired consequence of pre-snRNA sequences, which cells have to overcome. Our mutational analysis of U1, U2 and U4 snRNAs, when we weakened NSS indicated that destabilization of NSS does not affect snRNP biogenesis (Figs. 5-7), suggesting that the later variant is more likely. We did not identify NSS presence in minor snRNAs and we therefore did not analyze minor snRNAs any further. To clarify this issue we clearly state this fact in the text (p. 4, third paragraph).

Does NSS affect the binding of snRNAs to Gemin5?

We knocked down Gemin5 and did not observe any significant changes in U2 snRNP maturation, which indicates, that Gemin5 is not critical factor for snRNP biogenesis in human cultured cells. We therefore did not analyze a role of Gemin5 any further and focused on Gemin3, where we observed a clear phenotype upon the knockdown. Several structural studies showed that the WD40 domain of Gemin5 interacts with a stretch of uridines forming the Sm binding site plus adenine located at the 5' end of the Sm binding site (Xu et al. 2016 Genes&Dev; Jin et al, 2016 Genes&Dev; Tang et al. 2016, Cell Res.). Formation of NSS would sterically interfere with this mode of binding. However, Gemin5 also interacts with 7mG-cap found at the 5' end of pre-snRNA (Jin et al, 2016 Genes&Dev; Tang et al. 2016, Cell Res.), which together with the 3' terminal loop can represent additional contact points between snRNAs and Gemin5.

2. The authors produced weak and stable NSS U2 mutants. It is surprising to find that the stable NSS is less efficient in forming snRNPs and localizing to Cajal bodies compared to weak (or relaxed) NSS U2. In this scenario, what does Gemin 3 do? RNA helicases in general are supposed to destabilize helical structures. Stable NSS is a perfect match of Sm sequences which are mostly A-U base pairs. It is difficult to imagine that an RNA helicase cannot unwind this strength of dsRNA.

The reviewer is correct that a general RNA helicase should be able to unwind even stable NSS in stNSS variants of snRNAs tested in our work. However, Gemin3 belongs to a family of DEAD-box helicases that are in general non-productive and induce only partial relaxation of double stranded RNAs during RNP biogenesis (Gilman et al. 2017, Biochem Soc Trans). It is therefore conceivable that Gemin3 would not be able to unwind stNSS, which is consistent with our results (Fig. 5-7). To clarify this fact, we added a paragraph into Discussion to explain the function of DEAD-box helicases (p. 11, bottom paragraph).

3. Are NSSs in other snRNAs similar or are they different? Can authors compare the strength of NSS across snRNAs?

We calculated the strength of individual structural elements in each pre-snRNA. The data show that NSS and structures around the Sm site are among the least thermodynamically stable parts of pre-snRNA molecule (Fig S11). However, the strength of NSS within a particular pre-snRNA is strongly influenced by the stability of the whole structure. Therefore, comparing stability of NSS among different pre-snRNAs would have only very limited predictive value and we did not perform this comparison to avoid unsubstantiated conclusions.

4. The correlation between NSS and Gemin3 is not clear in the current study.

To address this reviewer's comment, we performed additional experiments to strengthen our conclusion that Gemin3 relaxes NSS. First, we created U1, U2 and U4 pre-snRNA variants where we inserted mutations that destabilize NSS (weak/noNSS mutants). In all these cases, Sm ring formation on noNSS constructs was Gemin3-independent indicating that artificial relaxation of NSS structure removes the requirement for Gemin3 action (Figs. 5, 7a and new Fig. 6). Second, we prepared a Gemin3-degron cell line, isolated the SMN complex lacking Gemin3 and showed that removal of Gemin3 reduced the ability of the SMN to relax U2 NSS (new Fig. 7c). Unfortunately, degradation of Gemin3 removed also Gemin4 and 5 from the SMN complex (Fig. S17), which has been reported previously (Otter et al. 2007, J. Biol. Chem.). However, together with RNAi experiments (Fig. 5), our data together strongly indicate that Gemin3 acts on NSS and is important during snRNP biogenesis to relax compact structure of pre-snRNA substrates.

Reviewer #3

We thank the reviewer for positive comments on our work and constructive suggestions that improved the manuscript.

Major Comments:

1. There is not enough experimental data to support some conclusions from the abstract (e.g., Gemin3 function). The abstract has to be rewritten, or new experimental data must be added (please see the comments below).

We added new experiments to support our conclusion about Gemin3 function. Namely, the relaxation of compacted structure in U1 and U4 pre-snRNA (noNSS constructs) removes the requirement for Gemin3 action during snRNP biogenesis. These experiments shown at new Fig. 6 are fully consistent with results acquired with U2 snRNA (weak/noNSS constructs, Fig. 5 and 7a). We also established a Gemin3-degron cell line, depleted cells of Gemin3 and isolated the SMN complex. This complex lacking Gemin3 (and also Gemin4 and 5 as reported previously (Otter et al. 2007, J. Biol. Chem.)) is less efficient in relaxation of the molecular beacon that mimics U2 snRNA NSS (new Fig. 7c). We believe that these new data further support our conclusions that Gemin3 is important for opening up compact pre-snRNA templates to allow Sm protein binding.

2. I am concerned that all the experimental results focus entirely on human U2 snRNA. The authors apply in silico modeling to predict suboptimal secondary structure models for four pre-snRNA/sn-RNA from 9-11 species, but only one model (for one species) was experimentally confirmed. It is a little strange that the authors, familiar with a very sophisticated method of RNA structure mapping (SHAPE-Map), performed the SHAPE-Map experiment only for human U2 snRNA. Although the authors found that in vitro and ex vivo SHAPE reactivities match predicted U2 snRNA structure, there is growing evidence that RNA models predicted in silico without experimental constraints are often inconsistent with experimentally determined structures (Rice et al. 2014, doi: 10.1261/rna.043323.113; Busan et al. 2019, doi: 10.1021/acs.biochem.8b01218, figure 3; Deigan et al. 2009, doi: 10.1073/pnas.0806929106, tables 1 and 2; Siegfried et al. 2014, doi: 10.1038/nmeth.3029. Epub 2014 Jul 13, figure 3.

The manuscript would benefit from showing mapping and functional results for other snRNAs.

To address the reviewer concern, we mapped secondary structures of in vitro transcribed U1 and U4 pre-snRNAs by SHAPE-MaP. The SHAPE-MaP analysis of U1 snRNA did not provide a clear outcome and the results were ambiguous and we were unable to draw any conclusion. There could be several reasons for this result and we discuss them in Discussion (p. 10, bottom paragraph and p. 11, top paragraph). U4 pre-snRNA SHAPE MaP analysis was consistent with the structural fold predicted by our in-silico modeling (new Fig. 2c). We also performed new functional analysis with U1 and U4 pre-snRNA (see next point).

3. Moreover, the new function of Gemin3 helicase was also showed just for human U2 snRNA.

We knocked down Gemin3 and analyzed biogenesis of U1 and U4 pre-snRNAs. We further introduced mutations that destabilize NSS in U1 and U4 pre-snRNAs and showed that these constructs do not require Gemin3 to mature. These data are now included in new Fig. 6 and are fully consistent with U2 snRNA results presented in previous version of the manuscript (Fig. 5).

Minor Required Revision:

1. Sm – please provide the full name of this protein, e.g., in the introduction.

Done. (p. 2, first paragraph of Introduction)

2. Page 2. "All spliceosomal snRNAs (except U6 and U6atac) are synthesized by RNA polymerase." Pls correct.

Corrected, thank you!

3. Page 3. Lines 18-20 It is accurate that a single MFE structure represents only one structure for a given sequence. However, the MFE algorithm also allows the prediction of multiple structures for one RNA sequence. Please clarify.

We did not mention the possibility of multiple MFE structures, as their occurrence is very rare and usually a single MFE structure is considered by most RNA secondary structure prediction algorithms. Nevertheless, we understand the reviewer's point and corrected the mentioned sentence to avoid the strong statement that MFE cannot predict more than one structure (p. 3, first paragraph of Results).

4. Page 5. Lines 34-39; I do not understand why SHAPE reactivities were mapped on in silico predicted structure or in vivo structure from another study. BTW Please, provide a source of in vivo model. Why did the authors not use the obtained SHAPE reactivities as pseudo-energy constraints for RNA structure modeling?

We modelled U2 snRNA structures (ex vivo and in vitro) using SHAPE reactivities as constraints as suggested by the reviewer and received very similar structures as using in silico modelling (compare Fig. 1b and Fig. 2a). The observed minor inconsistencies might result from inherited inaccuracies of both computational modeling and SHAPE experiments. For in vivo structure, we mapped SHAPE reactivities on the published structure from Dybkov et al. 2006 and included this reference in the text as proposed. In case of in vivo structure, we did not apply de novo prediction because protein binding prevents modification by the SHAPE reagent, which can affect modeling of the RNA structure. We changed the text (p. 5 bottom paragraph) and Figure 2 legend accordingly.

5. Page 6. Lines 1-4; It is pretty easy to test how Sm binding impacts SHAPE modifications using in vitro system similar to that described in the next chapter of the results section.

Actually, in vitro Sm ring assembly never works completely and the Sm ring is formed only on a subset of snRNA molecules. The reaction then contains a mixture of molecules with various SHAPE reactivities thus might not provide a clear result. We have no reason to doubt previous findings that protein binding prevents SHAPE reactivity and we therefore did not perform the SHAPE-MaP experiment with the in vitro assembled Sm ring.

6. Discussion. Page 9. Lines 9-11; I think that stem-loops I, III, and IV are stable. Please correct me if I am wrong.

You are absolutely correct, thank you!

7. Page 14.; Could you explain why RNA was folded in 57 mM MgCl₂? It is a very high concentration for this type of experiment.

First, we applied MgCl₂ concentration that we used for in vitro experiments with U2 snRNA and Sm ring formation. Next, we tested several MgCl₂ concentrations (0-60mM) for U4 pre-snRNA and 60mM MgCl₂ provided the best results with respect to snRNA folding.

8. Page 22.; Legend of Figure 2. "... were mapped onto mature (ex vivo) or...." Probably in vivo is correct. Yes, you are right, corrected.

Reviewers' Comments:

Reviewer #1:

Remarks to the Author:

Re-review of Panek et al.

Nat Comms ms. # NCOMMS-21-22591-A

In this revised manuscript, the authors have made a reasonable effort to respond to the critiques of the previous review, but there are still a few things that must be addressed. In paragraphs below I will respond to the responses of the authors.

Respectfully signed, Greg Matera

Reviewer 1

1. First of all, with the exception of human pre-snRNAs, it's not entirely clear if any of these metazoan 3'-extended precursors have been detected in vivo. We looked hard for them by northern blotting many years ago in flies and could not detect them. They could be short lived. Are the authors aware of any data along these lines in any of the other metazoans they analyze in Fig. 1? (This may be more of a minor point, but additional experimental evidence for the whole bioinformatic approach the authors take seems fundamental to me).

RESPONSE: The reviewer is correct that the data for pre-snRNAs from non-human species are missing. To overcome this issue, we estimated the length of 3' end extensions based on human results. We understand that this is only approximation. We therefore experimentally tested and validate the predicted structures for human snRNA by SHAPE-MaP. Because our main focus is put on human snRNAs, we did not experimentally validate other non-human snRNAs.

Critique: Without experimental evidence for (or against?) 3' extensions in any other species besides human, the authors will need to limit their conclusions. That means changes to the title, abstract and throughout the ms to qualify the results. For example:

Line 28 should be changed to read: "Human Sm-class snRNAs are generated as 3'-extended precursors that are exported to the cytoplasm, and assembled together with Sm proteins into core RNPs by the SMN complex. Here, we provide evidence that these pre-snRNA substrates contain compact, evolutionarily-conserved secondary structures that overlap with the Sm binding site. Structural motifs in these precursors are predicted to interfere with Sm core assembly. We model..."

Line 1: Title should be altered to say 'drives structural changes in metazoan snRNAs' ?

2. It is not clear from the manuscript or the supplemental figures and tables which species the authors consider as being a Protozoan. In looking over the supplementary table S1, it appears that *S.pombe* and *S.cerevisiae* have been lumped in with the protists. And on pg 9 the authors state something about "... in yeast and other protozoan U2 snRNAs." This is a huge mistake. And an opportunity (I hope!). A very large number of fungal genomes have been sequenced, and I believe the authors should re-do their folding analysis using fungi as a third group of organisms.

RESPONSE: Big thanks for pointing this out! We completely recalculate the pre-snRNA structures and applied in silico modeling on 15 fungi species and 14 protozoan species (Table S1). These data indicate that the structural motif around the Sm site is conserved only in metazoans (Fig. 1d). We further discuss the evolutionary aspect of NSS and Gemin3 appearance in evolution in Discussion (p. 12, first paragraph).

Critique: Ok, so IF the non-metazoan species WERE to express 3-extended precursor snRNAs, then they would not be predicted to form an NSS? I think the authors are missing a big opportunity here to

make a broader conclusion.

3. The big question is this: Do the predicted fungal pre-snRNAs have a base-paired NSS or is the Sm-site of for this class of organisms mostly single stranded? If fungal pre-snRNAs do not form an NSS, then that might help explain why Gemin3 and Gemin4 have become dispensable. Some in vitro SHAPE analysis or other structure probing to confirm this prediction would be important here as well.

RESPONSE: Our main aim in the first part of the manuscript was to determine structure of human pre-snRNAs. We applied an in-silico approach that relies on statistics (= identifying the most frequent structural folds) and comparison among species. The structural predictions of non-metazoan species were originally in the same group as metazoan, because we assumed that pre-snRNA structures are conserved among all eukaryotes. During the course of the work, we realized that metazoan pre-snRNAs stand out and have a conserved pre-snRNA structures while other species do not. We therefore sequestered metazoan and non-metazoan species and analyzed them separately. However, our main interest remained in human snRNAs and we therefore experimentally tested and validated the new predicted structures for human U2 snRNA and U4 pre-snRNAs (Fig. 2) and did not analyze other non-human samples.

Critique: From the authors response, it seems as if their in silico predictions simply fail to identify a predominant class of secondary structure in the various non-metazoan species. Given the inability to model the structure computationally, it seems like some sort of experimental structural analysis of an in vitro transcribed would be in order. Again, in the absence of experimental data for fungal or protozoan RNAs, the conclusions need to be limited to humans and other closely related organisms.

4. In Figures 5 and 6, the authors focus on RNAi experiments targeting Gemin3 and Gemin5 and they analyze the effects on nuclear import/Cajal body transport. Gemin3 knockdown has an effect but not Gemin5. The negative result for Gemin5 is fine, as this protein is really more of a separate subunit of its own. However, within the context of the SMN complex, Gemin3 has a dedicated binding partner in Gemin4.

Why have the authors not carried out RNAi on Gem4? Moreover, Gem4 is the only member of the SMN complex that actually contains a nuclear localization signal (NLS), and overexpression of this protein drives the SMN complex into the nucleus (see Meier et al. 2018). The authors should include Gem4 RNAi in the experiments shown in Figs 5 and 6. Note, there are published siRNAs for Gem4 that give good knockdown, so that's not an excuse (Shpargel 2005, PNAS).

RESPONSE: We have knockdown Gemin4 as suggested by the reviewer and observed a similar, albeit weaker, phenotype upon U2 snRNA microinjection. These data are included in Fig. 5a.

Critique: Ok, that seems reasonable.

Comments on responses to Reviewer 2

RESPONSE 1: The reviewer is absolutely correct that the role of NSS is not clear. We do not know, whether NSS has any particular role in snRNP biogenesis or whether its formation is just a by-product and undesired consequence of pre-snRNA sequences, which cells have to overcome. Our mutational analysis of U1, U2 and U4 snRNAs, when we weakened NSS indicated that destabilization of NSS does not affect snRNP biogenesis (Figs. 5-7), suggesting that the later variant is more likely. We did not identify NSS presence in minor snRNAs and we therefore did not analyze minor snRNAs any further. To clarify this issue we clearly state this fact in the text (p. 4, third paragraph).

Q: Do human minor-class snRNAs contain 3'-extended precursors? They might not.

Line 454: It should be noted that Sm core formation using a reconstituted SMN complex is not dependent on Gemin3 and ATP. The likely explanation is that the snRNA substrates prepared by in vitro transcription might adopt alternative structures and the reaction solution contains a mixture of differently folded snRNAs (see also different structural folds for U1 snRNA in Fig. 1).

Critique: Many of the studies using reconstituted SMN complex and in vitro transcribed snRNAs did not use 3'-extended precursors. So that might explain why Gemin3 not required in vitro. The authors correctly go on to point out that that Gemin3 becomes essential for Sm assembly in the presence of other factors present in nuclear extracts. That is to say, RNAi for Gem3 does interfere with Sm assembly in HeLa nuclear extracts (Shpargel 2005 and other refs).

Reviewer #2:

Remarks to the Author:

The authors improved the manuscript substantially and addressed all concerns.

Reviewer #3:

Remarks to the Author:

My major point of criticism concerned the limited amount of data supporting the conclusions. The current manuscript version contains additional interesting data that together with previous findings, make the manuscript much more valuable.

I really appreciate new experiments supporting the Gemin3 function.

Although new structural analyses (SHAPE-MaP) are moderate, the functional assays with structural U1 and U4 pre-snRNA are convincing.

I understand that Sm ring forming is not complete, but Fig 2c shows a clear reactivity decrease in Sm and a concurrent reactivity increase for a significant part of NSS for U2 in vivo. Similar analyses in diverse conditions should be performed for other pre-snRNA. Presentation for U4 only in vitro reactivity/model is not convincing. Alternatively, authors can use SHAPE-MaP for Gemin3/ pre-snRNA to directly confirm Gemin3 activity in NSS destabilization.

RESPONSE TO REVIEWERS' COMMENTS

Reviewer #1

1. Without experimental evidence for (or against?) 3' extensions in any other species besides human, the authors will need to limit their conclusions. That means changes to the title, abstract and throughout the ms to qualify the results. For example: Line 28 should be changed to read: "Human Sm-class snRNAs are generated as 3'-extended precursors that are exported to the cytoplasm, and assembled together with Sm proteins into core RNPs by the SMN complex. Here, we provide evidence that these pre-snRNA substrates contain compact, evolutionarily-conserved secondary structures that overlap with the Sm binding site. Structural motifs in these precursors are predicted to interfere with Sm core assembly. We model..."
Line 1: Title should be altered to say 'drives structural changes in metazoan snRNAs'

Our response: We followed the reviewer's suggestion and focused the text primarily on human snRNAs. Namely, we added "human" to the Title and modified Abstract and Results according to reviewer's suggestions.

2. Ok, so IF the non-metazoan species WERE to express 3-extended precursor snRNAs, then they would not be predicted to form an NSS? I think the authors are missing a big opportunity here to make a broader conclusion.

Our response: We computationally found structural evolutionary conservation for Metazoa, but not for Fungi and Protista. We therefore focused on humans, for which extended precursors have been experimentally determined. We strongly believe that extra sequences exist in other metazoans as well. Otherwise, it would not be possible to calculate structural conservation, as extra sequences play an essential role in the formation of pre-snRNA secondary structures and specifically in their structural rearrangements. On the other hand, we did not detect structural conservation for pre-snRNAs in Fungi and Protista. Thus, we decided against speculating on the existence of 3' extended precursors or their length in these classes of organisms. For this reason, we did not analyze or discuss them further. To clarify this point, we have included two sentences at the end of the first part of the Results (end of the first paragraph, p. 5) and in the Discussion (first paragraph, top of p. 10) that clarifies this point. We agree with the reviewer that potential existence of 3' extended precursors in non-metazoan species is interesting but it is out of the scope of our current manuscript.

3. From the authors response, it seems as if their in silico predictions simply fail to identify a predominant class of secondary structure in the various non-metazoan species. Given the inability to model the structure computationally, it seems like some sort of experimental structural analysis of an in vitro transcribed would be in order. Again, in the absence of experimental data for fungal or protozoan RNAs, the conclusions need to be limited to humans and other closely related organisms.

Our response: Again, we followed reviewer's suggestion and focused the text on human snRNAs (Title, Abstract, Results, Discussion). Please, see also our answer to the previous points for details.

4. Do human minor-class snRNAs contain 3'-extended precursors? They might not.

Our response: According to Young et al. 2010, Mol Cell 38:551-62, minor U11, U12 and U4atac also contain extra sequences.

5. Line 454: It should be noted that Sm core formation using a reconstituted SMN complex is not dependent on Gemin3 and ATP. The likely explanation is that the snRNA substrates prepared by in vitro transcription might adopt alternative structures and the reaction solution contains a mixture of differently folded snRNAs (see also different structural folds for U1 snRNA in Fig. 1).

Many of the studies using reconstituted SMN complex and in vitro transcribed snRNAs did not use 3'-extended precursors. So that might explain why Gemin3 not required in vitro. The authors correctly go on to point out that that Gemin3 becomes essential for Sm assembly in the presence of other factors present in nuclear extracts. That is to say, RNAi for Gem3 does interfere with Sm assembly in HeLa nuclear extracts (Shpargel 2005 and other refs).

Our response: We have included a sentence about in vitro experiments that do not use 3' extended precursors to assemble the Sm ring to further explain why Gemin3 (and Gemin4) is required only under more complex conditions (cell extracts, in cellulo). We also referred to Shpargel et al. 2005 and Almstead & Sarnow 2007, where Gemin3 knockout was shown to inhibit Sm ring formation (Discussion, p. 12, middle paragraph).

Reviewer #3

My major point of criticism concerned the limited amount of data supporting the conclusions. The current manuscript version contains additional interesting data that together with previous findings, make the manuscript much more valuable.

I really appreciate new experiments supporting the Gemin3 function.

Although new structural analyses (SHAPE-MaP) are moderate, the functional assays with structural U1 and U4 pre-snRNA are convincing.

I understand that Sm ring forming is not complete, but Fig 2c shows a clear reactivity decrease in Sm and a concurrent reactivity increase for a significant part of NSS for U2 in vivo. Similar analyses in diverse conditions should be performed for other pre-snRNA. Presentation for U4 only in vitro reactivity/model is not convincing..

Our response: We are pleased that the reviewer appreciated our functional assays with U1 and U4 snRNAs, which we have included in the revised version of the manuscript. And we would like to explain why we present SHAPE-MaP results with U2 snRNA (in vitro, in vivo and ex vivo) and U4 snRNA (in vitro only) and not the others snRNAs in the manuscript. We also applied SHAPE-MaP to map the structure of U1 snRNA under all conditions (in vivo, ex vivo and in vitro). The in vitro conditions did not provide a clear result, which prevented us from drawing a meaningful conclusion. The ex vivo and in vivo conditions confirmed previously published structures for mature U1 snRNAs, and we therefore chose not to report these data in the manuscript. See SHAPE-MaP results for U1 snRNA in Figure-for-review_3 - A. Pre-snRNA U5 was not functional in our assays and did not reach Cajal bodies after microinjection. Therefore, we did not analyze the pre-snRNA U5 further.

We also chose not to perform SHAPE-MaP for U4 snRNA under in vivo and ex vivo conditions. The main reason is that inside human cells almost all U4 snRNA is in complex with U6 snRNA (see Figure for reviewer_3 - B), which significantly alters its structure. Only a very limited amount of U4 snRNA is free, which we would not be able to detect using SHAPE-MaP. Because the U4/U6 snRNA is an irrelevant substrate for the SMN complex, we chose not to perform SHAPE-MaP for U4 snRNA under in vivo conditions. Similarly, deproteinization of the sample does not lead to melting of the U4/U6 helix. We would have to denature the RNA sample using high temperature. These conditions would not provide any new information compared to in vitro conditions. Therefore, we decided to devote our efforts to analyze U4 pre-snRNA in vitro using SHAPE-MaP. In this manuscript, we present two alternative approaches, SHAPE-MaP and mathematical modeling. Both approaches support nearly identical structural composition. Therefore, we believe that we have, with high probability, identified a structure that pre-U4 snRNA adopts.

Alternatively, authors can use SHAPE-MaP for Gemin3/ pre-snRNA to directly confirm Gemin3 activity in NSS destabilization.

Our response: We strongly believe that SHAPE-MaP is not ideal for monitoring Gemin3 NSS destabilization in vitro, and that other quantitative tools would be superior. For that reason, we used a U2-based molecular beacon with a fluorescent dye and quencher to monitor NSS destabilization by the SMN complex (Figure 7). We believe that these experiments can provide a similar information and replace the proposed SHAPE-MaP before and after Gemin3 treatment.

Supplementary figure for reviewer #3

A. SHAPE-MaP reactivities of U1 snRNA inside cells (*in vivo*) and U1 snRNA isolated from cells (*ex vivo*). The structures were model based on the reactivities. The grey bases were not covered by the amplicons and their structure was only modeled without experimental support. Red line indicates the Sm binding site. Note that Sm site is not reactive under *in vivo* condition likely due to Sm protein binding around the site. This creates an artificial stem-loop between nucleotides 118-134. In both cases, the general structural fold is consistent with published U1 snRNA structures.

B. Structure of U4/U6 based on Wersig & Bindereif 1990 (DOI: 10.1093/nar/18.1.6223) and Charenton et al. 2019 (DOI: 10.1126/science.aax3289). The extensive base-pairing between U4 and U6 snRNAs is indicated.

Reviewers' Comments:

Reviewer #1:

Remarks to the Author:

Re-review of Panek et al.

Nat Comms ms. # NCOMMS-21-22591-A

In this revised manuscript, the authors have made a reasonable effort to respond to the critiques of the previous review, but there are still a few things that must be addressed. In paragraphs below I will respond to the responses of the authors.

Respectfully signed, Greg Matera

Reviewer 1

1. First of all, with the exception of human pre-snRNAs, it's not entirely clear if any of these metazoan 3'-extended precursors have been detected in vivo. We looked hard for them by northern blotting many years ago in flies and could not detect them. They could be short lived. Are the authors aware of any data along these lines in any of the other metazoans they analyze in Fig. 1? (This may be more of a minor point, but additional experimental evidence for the whole bioinformatic approach the authors take seems fundamental to me).

RESPONSE: The reviewer is correct that the data for pre-snRNAs from non-human species are missing. To overcome this issue, we estimated the length of 3' end extensions based on human results. We understand that this is only approximation. We therefore experimentally tested and validate the predicted structures for human snRNA by SHAPE-MaP. Because our main focus is put on human snRNAs, we did not experimentally validate other non-human snRNAs.

Critique: Without experimental evidence for (or against?) 3' extensions in any other species besides human, the authors will need to limit their conclusions. That means changes to the title, abstract and throughout the ms to qualify the results. For example:

Line 28 should be changed to read: "Human Sm-class snRNAs are generated as 3'-extended precursors that are exported to the cytoplasm, and assembled together with Sm proteins into core RNPs by the SMN complex. Here, we provide evidence that these pre-snRNA substrates contain compact, evolutionarily-conserved secondary structures that overlap with the Sm binding site. Structural motifs in these precursors are predicted to interfere with Sm core assembly. We model..."

Line 1: Title should be altered to say 'drives structural changes in metazoan snRNAs' ?

2. It is not clear from the manuscript or the supplemental figures and tables which species the authors consider as being a Protozoan. In looking over the supplementary table S1, it appears that *S.pombe* and *S.cerevisiae* have been lumped in with the protists. And on pg 9 the authors state something about "... in yeast and other protozoan U2 snRNAs." This is a huge mistake. And an opportunity (I hope!). A very large number of fungal genomes have been sequenced, and I believe the authors should re-do their folding analysis using fungi as a third group of organisms.

RESPONSE: Big thanks for pointing this out! We completely recalculate the pre-snRNA structures and applied in silico modeling on 15 fungi species and 14 protozoan species (Table S1). These data indicate that the structural motif around the Sm site is conserved only in metazoans (Fig. 1d). We further discuss the evolutionary aspect of NSS and Gemin3 appearance in evolution in Discussion (p. 12, first paragraph).

Critique: Ok, so IF the non-metazoan species WERE to express 3-extended precursor snRNAs, then they would not be predicted to form an NSS? I think the authors are missing a big opportunity here to

make a broader conclusion.

3. The big question is this: Do the predicted fungal pre-snRNAs have a base-paired NSS or is the Sm-site of for this class of organisms mostly single stranded? If fungal pre-snRNAs do not form an NSS, then that might help explain why Gemin3 and Gemin4 have become dispensable. Some in vitro SHAPE analysis or other structure probing to confirm this prediction would be important here as well.

RESPONSE: Our main aim in the first part of the manuscript was to determine structure of human pre-snRNAs. We applied an in-silico approach that relies on statistics (= identifying the most frequent structural folds) and comparison among species. The structural predictions of non-metazoan species were originally in the same group as metazoan, because we assumed that pre-snRNA structures are conserved among all eukaryotes. During the course of the work, we realized that metazoan pre-snRNAs stand out and have a conserved pre-snRNA structures while other species do not. We therefore sequestered metazoan and non-metazoan species and analyzed them separately. However, our main interest remained in human snRNAs and we therefore experimentally tested and validated the new predicted structures for human U2 snRNA and U4 pre-snRNAs (Fig. 2) and did not analyze other non-human samples.

Critique: From the authors response, it seems as if their in silico predictions simply fail to identify a predominant class of secondary structure in the various non-metazoan species. Given the inability to model the structure computationally, it seems like some sort of experimental structural analysis of an in vitro transcribed would be in order. Again, in the absence of experimental data for fungal or protozoan RNAs, the conclusions need to be limited to humans and other closely related organisms.

4. In Figures 5 and 6, the authors focus on RNAi experiments targeting Gemin3 and Gemin5 and they analyze the effects on nuclear import/Cajal body transport. Gemin3 knockdown has an effect but not Gemin5. The negative result for Gemin5 is fine, as this protein is really more of a separate subunit of its own. However, within the context of the SMN complex, Gemin3 has a dedicated binding partner in Gemin4.

Why have the authors not carried out RNAi on Gem4? Moreover, Gem4 is the only member of the SMN complex that actually contains a nuclear localization signal (NLS), and overexpression of this protein drives the SMN complex into the nucleus (see Meier et al. 2018). The authors should include Gem4 RNAi in the experiments shown in Figs 5 and 6. Note, there are published siRNAs for Gem4 that give good knockdown, so that's not an excuse (Shpargel 2005, PNAS).

RESPONSE: We have knockdown Gemin4 as suggested by the reviewer and observed a similar, albeit weaker, phenotype upon U2 snRNA microinjection. These data are included in Fig. 5a.

Critique: Ok, that seems reasonable.

Comments on responses to Reviewer 2

RESPONSE 1: The reviewer is absolutely correct that the role of NSS is not clear. We do not know, whether NSS has any particular role in snRNP biogenesis or whether its formation is just a by-product and undesired consequence of pre-snRNA sequences, which cells have to overcome. Our mutational analysis of U1, U2 and U4 snRNAs, when we weakened NSS indicated that destabilization of NSS does not affect snRNP biogenesis (Figs. 5-7), suggesting that the later variant is more likely. We did not identify NSS presence in minor snRNAs and we therefore did not analyze minor snRNAs any further. To clarify this issue we clearly state this fact in the text (p. 4, third paragraph).

Q: Do human minor-class snRNAs contain 3'-extended precursors? They might not.

Line 454: It should be noted that Sm core formation using a reconstituted SMN complex is not dependent on Gemin3 and ATP. The likely explanation is that the snRNA substrates prepared by in vitro transcription might adopt alternative structures and the reaction solution contains a mixture of differently folded snRNAs (see also different structural folds for U1 snRNA in Fig. 1).

Critique: Many of the studies using reconstituted SMN complex and in vitro transcribed snRNAs did not use 3'-extended precursors. So that might explain why Gemin3 not required in vitro. The authors correctly go on to point out that that Gemin3 becomes essential for Sm assembly in the presence of other factors present in nuclear extracts. That is to say, RNAi for Gem3 does interfere with Sm assembly in HeLa nuclear extracts (Shpargel 2005 and other refs).

Reviewer #2:

Remarks to the Author:

The authors improved the manuscript substantially and addressed all concerns.

Reviewer #3:

Remarks to the Author:

My major point of criticism concerned the limited amount of data supporting the conclusions. The current manuscript version contains additional interesting data that together with previous findings, make the manuscript much more valuable.

I really appreciate new experiments supporting the Gemin3 function.

Although new structural analyses (SHAPE-MaP) are moderate, the functional assays with structural U1 and U4 pre-snRNA are convincing.

I understand that Sm ring forming is not complete, but Fig 2c shows a clear reactivity decrease in Sm and a concurrent reactivity increase for a significant part of NSS for U2 in vivo. Similar analyses in diverse conditions should be performed for other pre-snRNA. Presentation for U4 only in vitro reactivity/model is not convincing. Alternatively, authors can use SHAPE-MaP for Gemin3/ pre-snRNA to directly confirm Gemin3 activity in NSS destabilization.